# Reduction of Prostate Cancer Risk: Role of Frequent Ejaculation-Associated Mechanisms

**DOI:** 10.3390/cancers17050843

**Published:** 2025-02-28

**Authors:** Mohamed Hassan, Thomas W. Flanagan, Abdulaziz M. Eshaq, Osama K. Altamimi, Hassan Altalag, Mohamed Alsharif, Nouf Alshammari, Tamadhir Alkhalidi, Abdelhadi Boulifa, Siraj M. El Jamal, Youssef Haikel, Mossad Megahed

**Affiliations:** 1Institut National de la Santé et de la Recherche Médicale, University of Strasbourg, 67000 Strasbourg, France; youssef.haikel@unistra.fr; 2Department of Operative Dentistry and Endodontics, Dental Faculty, University of Strasbourg, 67000 Strasbourg, France; 3Research Laboratory of Surgery-Oncology, Department of Surgery, Tulane University School of Medicine, New Orleans, LA 70112, USA; eshaq@gwu.edu; 4Department of Pharmacology and Experimental Therapeutics, LSU Health Sciences Center, New Orleans, LA 70112, USA; tflan1@lsuhsc.edu; 5Department of Epidemiology and Biostatistics, Milken Institute School of Public Health, George Washington University, Washington, DC 20052, USA; 6College of Medicine, Alfaisal University, Riyadh 11533, Saudi Arabia; osaltamimi@alfaisal.edu (O.K.A.); hassanalitalag@gmail.com (H.A.); moalsharif@alfaisal.edu (M.A.); nalshammari@alfaisal.edu (N.A.); talkhaldii@outlook.com (T.A.); 7Berlin Institute of Health, Charité University Hospital, 10117 Berlin, Germany; boulifaabdelhadii@gmail.com; 8Competence Center of Immuno-Oncology and Translational Cell Therapy (KITZ), Charité-University Hospital, 10117 Berlin, Germany; 9Department of Pathology and Laboratory Medicine, Rutgers Robert Wood Johnson Medical School, New Brunswick, NJ 08901, USA; sml362@rwjms.rutgers.edu; 10Pôle de Médecine et Chirurgie Bucco-Dentaire, Hôpital Civil, Hôpitaux Universitaire de Strasbourg, 67000 Strasbourg, France; 11Clinic of Dermatology, University Hospital of Aachen, 52074 Aachen, Germany; mmegahed@ukaachen.de

**Keywords:** prostate cancer, testosterone, 5α-reductase-2, 5α-dihydrotestosterone

## Abstract

There is considerable evidence to suggest that frequent ejaculation reduces the risk of prostate cancer. The more frequently ejaculation occurs without risky sexual behavior, the lower the risk of prostate cancer. The mechanisms regulating the reduction in PCa risk associated with frequent ejaculations are attributed to the suppression of the sympathetic nervous system, resulting in a relief of tension and a slowing of the division of prostate epithelial cells. It is suggested that ejaculation frequency affects gene expression in prostate tissue and subsequently influences the susceptibility of the tissue to tumor formation, leading to a reduction in the risk of PCa.

## 1. Introduction

Prostate cancer (PCa) accounts for ~15% of diagnosed cancers among men [1,2]. Although the number of PCa-related deaths has decreased over the last decade [3,4], the incidence of this disease is increasing continuously worldwide [5,6].

The prostate is a small gland that is anatomically located below the bladder and in front of the rectum [7]. The main function of this organ is to produce fluids in semen and to push semen through the urethra during ejaculation, as well as to control the local and systemic concentrations of 5α-dihydrotestosterone [8,9]. 5α-Dihydrotestosterone is one of the stronger androgens expressed in prostate tissue [10,11]. The production of 5α-dihydrotestosterone in the prostate is regulated by the isoenzyme 5α-reductase-2, which leads to the conversion of precursor testosterone into 5α-dihydrotestosterone [12]. Although age-dependent enlargement of the prostate is a normal anatomical feature [13,14], the occurrence of PCa is age-dependent, occurring mostly in older men [15].

In addition to non-modifiable risk factors such as age, family history and ethnicity, modifiable risk factors include lifestyle choices such as diet, tobacco and alcohol consumption, obesity and physical inactivity, as well as exposure to environmental factors such as harmful chemicals and ionizing radiation play major roles in the development of PCa [2,16]. In contrast to the non-modifiable factors, modifiable factors such as diet and physical activity have a significant impact on reducing PCa risk [17,18].

The formation and homeostasis of the adult prostate are controlled by androgens, male hormones originating in the testes [19,20].

Androgens act by binding to the androgen receptor (AR), a hormonal transcription factor found in both the epithelium and stroma of the prostate, the activation of which is essential in translocation of androgens from the cytoplasm to the nucleus [21,22].

In addition to regulation of reproductive aspects, the glandular function of the prostate includes control of the local and systemic concentrations of androgens [19,23]. The main functions of androgens are to regulate the development, growth, and maintenance of the male genital system both in embryogenesis and adulthood [24,25]. Androgens likewise contribute to the development and progression of benign prostatic hyperplasia (BPH) and cancer [26,27]. Development of PCa is attributed to deregulation of the AR pathway [28,29]. Testosterone, the most abundant circulating androgen, is converted to 5α-dihydrotestosterone via the isoenzyme 5α-reductase-2 which is expressed in prostate tissue [12,30]. Although testosterone and 5α-dihydrotestosterone serve as natural ligands of the AR [22,31], the binding affinity of 5α-dihydrotestosterone to the androgen receptor is greater than that of testosterone [32,33]. Despite similar association rates of testosterone and 5α-dihydrotestosterone, the dissociation rate of 5α-dihydrotestosterone from AR is slower than that of testosterone [28,34]. Accordingly, the 5α-dihydrotestosterone-AR complex is expected to be more stable and active and play an important role in the maturation and physiology of male genitalia [28,33,35]. Animal models lacking 5α-reductase enzymes or humans with a mutation in the 5α-reductase gene can demonstrate underdeveloped male or fully developed female genitalia [28,36,37,38]. Males with attenuated 5α-dihydrotestosterone exhibit sexual dysfunction such as impaired erection, loss of libido, and reduced ejaculate volume [39,40]. Loss or reduction in 5α-dihydrotestosterone correlates significantly with PCa mortality [41,42]. Thus, frequent ejaculations and subsequently increased 5α-dihydrotestosterone levels may have an impact on reducing PCa risk [18,43]. Although the association between frequent ejaculation and reduction of PCa risk has been reported, the mechanisms regulating the frequent ejaculation-mediated reduction of PCa remain to be explored. This review sheds light on the mechanistic role of frequent ejaculation in reducing PCa risk.

## 2. Anatomy, Physiological Functions and Secretion of Prostate

The human prostate is one of the male accessory sex glands located below the bladder and in front of the rectum. It consists of five lobes, including an anterior lobe, a posterior lobe, two lateral lobes, and a middle lobe. The prostate structure consists of connective and glandular tissue covered by prostatic fascia, a layer of stretchable connective tissue [44,45]. The ejaculatory ducts carry the secretions of the seminal vesicles (fluid and sperm), pass from the ductus deferens through the cranial part of the gland, and open into the urethra at the verumontanum [44,46]. In addition to supplying fluid to the semen, the prostate is also responsible through its muscular function for forcing the sperm into the urethra and expelling them during ejaculation [44,45]. The functional structure of the prostate gland is shown in Figure 1.

As an additional sexual organ, the prostate secretes fluid into the semen, which supports sperm release and fertility [8,46]. The cells of the prostate can be roughly divided into two cell types: the parenchymal epithelial cells that form the acini and ducts of the gland and secrete prostatic fluid, and the stromal cells, which surround the acini and ducts and provide support and structure to the gland [46,47].

The main function of the accessory sex tissues, which include the prostate, seminal vesicles and urethral glands, is the production of seminal plasma, the non-gametic portion of semen [48,49]. The main function of the seminal plasma is to serve as a buffered, nutrient-rich transport medium for sperm when they are deposited in the female genital tract [50,51]. Seminal plasma is characterized by its alkalinity, which is essential to neutralize the acidic environment of the vagina [52,53]. The main function of the prostaglandins which are present in seminal plasma is to stimulate the contraction of smooth muscles along the genital tract of the female to facilitate the transport of the sperm to the egg [54,55]. In addition to zinc and IgA that are characterized by their bacteriostatic properties, the seminal plasm contains many prostatic proteins that are essential to prevent autoagglutination of sperm [56,57].

As is known, in healthy men, seminal plasma production, including the prostatic component, is regulated by circulating androgens via a hormonal control dependency [19,58]. Once circulating testosterone is transported into the prostate cells, it is converted by 5α-reductase into dihydrotestosterone (DHT), which in turn binds the androgen receptor (AR), dimerizes, and translates into the cell nucleus. The nuclear AR then binds the response elements of androgen-regulated genes to trigger gene expression of secretory proteins or enzymes to be secreted into the semen [19,39].

In addition to hormonal control, the prostatic secretions also underlay neural control [19,59]. It is noteworthy that the components of the prostate including both epithelial- and stromal-derived tissues are innervated by the autonomic nervous system [19,60].

As has been widely reported and documented, prostate fluid is rich in enzymes, proteins and minerals that are essential for the protection and nutrition of sperm [60,61]. Due to sexual arousal, the prostate can push prostate fluid through the ducts into the urethra, where it mixes with sperm and other fluids and is then ejaculated as semen [62,63]. The most common prostatic fluid components include zinc, citric acid, sodium, potassium, calcium, magnesium and chloride, spermine, prostaglandins, cholesterol and lipids, immunoglobulins, acid phosphatase, and prostate-specific antigen (PSA) as well as extracellular vesicles [64,65].

The highest concentration of zinc has been reported in the prostate gland, when compared with the other organs of the human body [64,65]. Therefore, prostatic secretions are the main source of the high concentration of zinc in the seminal plasma, which is estimated to be nearly 100-fold higher than that found in serum [66,67]. In addition to its function as an antibacterial factor both in seminal plasma and in non-ejaculatory prostate secretions, zinc is also thought to play an important role in the development of normal testes, spermatogenesis, and sperm motility [68]. Furthermore, the zinc in seminal plasma has been reported to function as a cell membrane and nuclear chromatin stabilizer in spermatozoa [69].

The binding of zinc to semenogelin I and II in seminal plasma is also essential for the regulation of semen liquefaction [70]. Semenogelin is a protein reported to be involved in the formation of a gel matrix that coats ejaculated sperm to prevent capacitation [71,72]. Citric acid is another of the secretory products of the prostate, which is characterized by its ability to bind metal ions. It is also one of the main anions in human seminal plasma [73,74]. Although higher concentrations of calcium and magnesium have been found in seminal plasma compared to other bodily fluids, the contribution of the prostate to the increase in these anions has not been reported [75,76]. However, variations in sodium, potassium, calcium, magnesium and chloride concentrations in prostate fluid have been reported to be closely associated with health or disease status [65,77]. This is partly a passive response to varying amounts of citrate secretion in the prostate fluid in the diseased state.

Spermine is one of the polyamines that is found in semen [78,79]. Spermine is one of the components of semen and its concentration in the prostate gland is the highest in any tissue in the human body [80,81]. It is also the main source of polyamine in seminal plasma [82,83]. The enzyme ornithine decarboxylase (ODC) is characterized by its ability to serve as a rate-limiting enzyme in the synthesis of polyamines and spermine [84]. ODC expression is increased in benign prostatic hyperplasia (BPH) tissue, suggesting a possible role of polyamines in the pathogenesis of this disease [85,86].

Although prostaglandins are present throughout the body, the highest concentrations of prostaglandins in the human body have been found in the seminal vesicles [87,88]. Prostaglandins are characterized by their potent physiological effects on men, particularly those involved in erection, ejaculation, sperm motility, and contractions of the testes and penis [89,90]. In addition, the deposition of prostaglandins in seminal fluid in the vagina has been reported to affect cervical mucus, vaginal secretions and uterine contractions [91].

The main source of the lipid fraction in seminal plasma comes from the prostate [92,93]. The most common components of the lipid fraction in semen include cholesterol and phospholipids [94]. These phospholipids are composed of 44% sphingomyelin, 12.3% ethanolamine plasmalogen and 11.2% phosphatidylserine [95,96]. The ratio of cholesterol to phospholipids is considered important to protect sperm from temperature and environmental shocks [97,98].

Immunoglobulins, including IgG and IgA as well as IgM, are present in seminal plasma at lower concentrations than those found in blood [99,100].

IgM concentration has been found to be lower in seminal plasma, when compared to that of IgG and IgA [101,102].

Acid phosphatase activity is one of the components of seminal fluid which is found in higher concentrations in the prostate than in any other tissue [103,104,105]. Prostatic acid phosphatase is a glycoprotein dimer, the substrate of which in the seminal plasma is phosphorylcholine phosphate which can be rapidly hydrolyzed by acid phosphatase [105,106]. Overexpression of acid phosphatase is observed in cancerous prostate tissues. For example, about 95% of patients with prostate cancer are characterized by high expression of acid phosphatase, especially in patients with metastases in the bones [106,107]. Accordingly, acid phosphatase is considered an important serum marker for screening and staging of prostate cancer, even if it is replaced by the more specific prostate-specific antigen (PSA).

PSA is a glycoprotein that acts as a serine protease and is produced almost exclusively in epithelial cells of the prostate [108,109]. PSA is involved in the dissolution of the seminal clot and uses semenogelin as a substrate [110]. The clinical utilization of PSA as an important serum marker for prostate cancer has been established [105,111]. PSA is an organ-specific rather than cancer-specific marker that is present in the serum of patients with BPH or prostatitis [112,113].

As part of the composition of seminal fluid, extracellular vesicles (EVs) are lipid bilayer nanovesicles, typically ranging in size from 30 to 1000 nm in diameter [114]. EVs serve as transporters of a wide range of molecules, including proteins, nucleic acids, metabolites, and lipids [115,116]. Seminal plasma contains comparatively more EVs compared to the levels circulating in blood or cerebrospinal fluid [114,115].

The main source of EVs in semen are the cells of the testes, epididymis, and vas deferens, as well as the cells of the male accessory glands, including the prostate and seminal vesicles [115]. They resemble EVs identified in other bodily fluids and constitute a heterogeneous population characterized by their variable size, shape and composition [116,117]. Semen EVs have the potential to bind to and exchange active molecules with mature sperm and endometrial epithelial cells and are thereby involved in the regulation of several reproductive processes [115]. Furthermore, it has been reported that the cargo load of sEVs differs between normozoospermic and non-normozoospermic men [118]. Seminal EVs may also facilitate the safe transit of spermatozoa in the female genital tract by regulating the uterine immune response [49,115]. Taken together, these findings suggest that semen EVs may be involved in fertility and that some biomolecules encapsulated in EVs could be considered as biomarkers of male fertility. However, among the molecules encapsulated in sEVs, none are currently recognized as biomarkers for male fertility. To determine how far science has advanced in this field, the present systematic review aims to summarize existing published evidence on the association of EVs with fertility and to identify candidate EV molecules that could function as biomarkers of fertility or infertility.

Prostasomes are EVs known as macrovesicles which can be secreted by the acinar epithelial cells of the prostate gland [57,119]. Prostasomes are intracellular vesicles that are mostly found inside large storage vesicles. Intracellular macrovesicles are similar to the multivesicular bodies of the late endosomal origin [57]. Therefore, it was assumed that prostasomes play an important role in intercellular communication, especially by directly interacting between the immotile acinar cells of the prostate and the motile sperm. In addition to the transfer of membrane components, prostasomes can also transfer genetic material to sperm [120]. Once the proteases are released from the secretory epithelial cells of the prostate gland and become part of the sperm, they can interact with the sperm membrane and eventually fuse with it [57,119]; thus, transfer of molecules from prostatic secretory cells to the spermatozoon has been reported to be mediated by prostasomes [57,119].

Accumulated evidence suggests activation of the renin–angiotensin system (RAS) in the human prostate [121,122,123]. The expression and cellular localization of angiotensin II (Ang II) and AT1 receptor proteins were analyzed in normal human prostate and benign prostatic hyperplasia (BPH). As has been reported, increased expression of Ang II and significant reduction of the AT1 receptor are common in prostate hyperplasia [121,122,123]. Whereas in the normal prostate, immunoreactivity is restricted to the basal layer of the epithelium, the AT1 receptor is predominantly localized in both the stroma and the smooth muscle of the prostatic vessels [121,122]. The downregulation of the AT1 receptor together with increased tissue levels of Ang II, as well as increased angiotensin-converting enzyme (ACE) at both mRNA and protein levels in BPH [124,125], suggests that RAS activation in BPH depends on its cellular localization and function. The role of Ang II and AT1 receptors in the regulation of prostate cancer has been demonstrated in several studies [126,127]. The observation suggests clinical relevance to inhibition of RAS activation in the treatment and prevention of PCa [128,129,130]. Several studies have also shown that angiotensin II receptor blockers (ARBs) have the potential to reduce elevated PSA levels in patients without PCa and significantly reduce the risk of PCa [130,131].

## 3. Regulation of Ejaculation

Ejaculation occurrence requires the synergistic activation of the autonomic and somatic nervous systems, as well as the contribution of spinal cord-dependent mechanisms [132,133]. Non-adrenergic, non-cholinergic innervation which modulates the activity of the accessory sex glands also modulates ejaculatory regulation [134,135]. Ejaculation generation, together with autonomic and somatic spinal nuclei, is controlled by a spinal network that determines stimulatory or inhibitory genital sensory and supraspinal inputs [133,134]. Likewise, a brain circuit responsible for ejaculation has been described [136,137]. This brain circuit is part of a larger network that controls other aspects of the sexual response [136,137]. The circuits include discrete neuronal populations distributed across all areas of the brain [138,139]. This expanded central nervous system network can diversify the neurotransmitter systems involved in the ejaculation process [132,140].

Neurotransmitters are chemical molecules that allow neurons to communicate with one another and transmit messages to muscles throughout the body [141,142]. Neurotransmitters functionally enable the brain to process chemical synaptic transmissions [143]. Neurotransmitters include those affecting the peripheral nervous system, the spinal cord, and neurotransmitters associated with the brain [141,142,143].

### 3.1. Neurotransmitters Associated with the Peripheral Nervous System

This group of neurotransmitters includes noradrenaline and acetylcholine, nitric oxide, oxytocin, purines and sensory receptors.

#### 3.1.1. Noradrenaline and Acetylcholine

The dense autonomic innervation of the male genitalia is a key mechanism that controls peripheral events leading to ejaculation [144,145]. Sympathomimetics are characterized by their ability to trigger contractions of the sperm ducts and sex glands [145,146]. This contractile mechanism is primarily mediated by α1 adrenoreceptors expressed by smooth muscle cells [147,148]. The induction of sex gland contractions by cholinomimetics likely occurs through muscarinic receptor activation-dependent mechanisms [143,144]. In addition to inducing gonad and sex gland contraction, stimulation of muscarinic receptors induces the secretion of seminal fluid from the seminal vesicles [144,149].

#### 3.1.2. Nitric Oxide

Nitric oxide-releasing fibers have been reported to form the main constituent of the non-cholinergic autonomic system in addition to being a main component of the male seminal tract [144,150]. Thus, the role of nitic oxide-releasing fibers in the regulation of contractile activity of the male seminal tract has been considered. Consequently, studies on the contractile function of nitric oxide-releasing fibers revealed that stimulation of the intracellular nitric oxide signaling pathway in the seminal vesicles is associated with a reduction in the contractile activity of smooth muscle [151,152]. Also, nitric oxide reduces the contractile activity of smooth muscle as well as inhibiting activity of the androgen receptor [153,154] and therefore inhibits prostate cancer growth [155,156].

#### 3.1.3. Peripheral Nervous System-Associated Oxytocin

Oxytocin is a hormone produced mainly by the pituitary gland and to a lesser extent the testis and prostate [157,158]. Oxytocin not only induces the contraction of smooth muscle cells in the genital tract through both direct and indirect mechanisms but also promotes sperm transport in the vas deferens [152,157]. Serum levels of oxytocin in men have been reported to correlate positively with ejaculation [159,160]. However, it is still not clear whether the increase of oxytocin release is the cause or the consequence of ejaculation [161,162]. While oxytocin is primarily a female hormone associated with breastfeeding and childbirth [163,164], the detection of this peptide within male reproductive tracts indicates a link between the release of oxytocin and ejaculation [165,166].

#### 3.1.4. Purines

Purinergic receptor subtype 2 (P2R) is highly expressed in smooth muscle cells of the genital tract of various species, including humans [167,168]. P2R is involved in the regulation of the pro-contractile effect of ATP in the vas deferens [167,168]. The purinergic system, which includes ATP and adenosine signaling pathways, has been shown to play an important role in the development and progression of PCa [169,170]. Activation of purinergic receptors, such as P2R, enhances the proliferation, migration, and invasion of PCa cells [171,172], with some studies indicating that ATP can inhibit PCa cell growth via P2R-dependent mechanism [173,174]. Consistently, the antineoplastic effects of purines are thought to be mediated by extracellular nucleotides and ectonucleotides in various tumor types [175,176].

#### 3.1.5. Peripheral Nervous System-Associated Serotonin

Serotonin, or 5-hydroxytryptamine (5-HT), is a monoamine neurotransmitter involved in the regulation of various functions, such as mood, as well as in the modulation of numerous physiological processes [177,178]. Serotonin is produced by various cell types including central nervous system (CNS) cell types, Merkel cells, neuroendocrine lung cells, and taste receptor cells of the tongue [179,180]. The 5-HT receptors represent a group of receptors, which differ greatly in their structures. These receptors are characterized by their distinctive intracellular properties and are divided into seven different families (5-HT1 to 5-HT7) [181]. As is known, all 5-HT receptors belong to the group of G protein-coupled receptors, except for the 5-HT3 receptor channel which is ligand-gated. 5-H receptors are common in the central and peripheral nervous systems, and their expression has been reported in almost all animals. The modulation of both excitatory and inhibitory neurotransmission is the main function of the 5-HT receptors [182]. 5-HT1A receptors are in the dorsal horn and therefore probably play a role in regulating ejaculatory mechanisms [183,184]. Accordingly, the location of 5-HT1A receptors in the intermediolateral nucleus (IML) of the lumbar and sacral segments suggests that these receptors may be important for the regulation of both sympathetic and preganglionic efferent activity [185,186].

Unlike 5-HT1A receptors, which have a somato-dendritic location on 5-HT neurons, the 5-HT1B receptors are located at terminals of 5-HT cells and play a role as auto-receptors that control 5-HT release in the synaptic cleft [187,188]. Selective 5-HT1B antagonists have been shown to prevent negative 5-HT feedback through this site and thus increase extraneuronal 5-HT [189,190]. The presence of 5-HT1B receptors in the spinal cord has been reported in several studies located at 5-HT terminals [191,192,193].

Accumulated evidence indicates that the overall effect of 5-HT on ejaculation is inhibitory [194,195]. The role of 5-HT and the 5-HT system in sexual reflex was considered, based on data obtained from an experimental model with anesthetized rats, as a spinal ejection reflex [196]. The release of 5-HT in spinal cord segments L3–L5 from the terminals of axons descending from the rostral region of the paragigantocellular nucleus (nPGi) has been shown to have an inhibitory effect on ejaculation [197,198]. The inhibitory influence of 5-HT projections originating from nPGi and terminating in lumbosacral segments was further supported in a recent study using another experimental paradigm that also mimics the ejaculatory reflex [199].

### 3.2. Spinal Cord-Linked Neurotransmitters

This group of neurotransmitters includes GABA, oxytocin, serotonin, and substance P.

#### 3.2.1. GABA

Because of the ubiquity of the c-aminobutyric acid system (GABA) in the central nervous system, GABA influences virtually all centrally controlled neurophysiological processes [200]. For example, administration of the GABA-B receptor subtype antagonist baclofen into the subarachnoid space alleviated refractory spasticity in patients with spinal cord injuries and in patients with multiple sclerosis as well [200,201,202,203,204]. Clinical observations have revealed that intrathecal injection of baclofen causes sexual dysfunction in men, as evidenced by loss of ejaculation, thus suggesting a possible role for the GABA-B antagonist in the regulation of male sexual function [205,206]. The likely molecular action responsible for this reduced sexual function and ejaculation is attributed to the inhibitory effect of GABA-B receptors on the spinal cord [207], and more specifically to the lumbosacral spinal cord on presynaptic sacral afferent reflexes [202,208]. Because baclofen acts on these receptors, the inhibition of reflex reactions in the penis is expected. This inhibitory relationship between GABA and sexual behavior in male rats can be explained by the dramatic increases in cerebrospinal fluid concentrations of GABA observed immediately after ejaculation, a time when male rats are unresponsive to sexual stimulation [209]. Examination of the copulatory behavior of male rats revealed that intrathecal baclofen had a much lower inhibitory effect than anticipated, as evidenced by high GABA levels noted in the cerebrospinal fluid after copulation [203,204,206].

#### 3.2.2. Spinal Cord-Linked Oxytocin

Oxytocin (OT) is a hypothalamic neuropeptide that has the potential to mediate several key neurophysiological functions associated with human behaviors and social interactions, including sexual arousal. However, the production of OT in the supraoptic (SON) and paraventricular (PVN) nuclei of the hypothalamus is regulated by both magnocellular and parvocellular neurons. Both magnocellular and parvocellular neurons are recognized as large cells. Thus, once the OT has been synthesized in the hypothalamus, the synthesized OT can then be transported within the axons to the posterior pituitary to be released into the bloodstream [210,211]. An intrathecally administered antagonist of oxytocin receptors was found to alter the ejaculatory response in a model of pharmacologically induced ejaculation [134,212]. It is therefore postulated that descending oxytocinergic projections from the paraventricular nucleus of the hypothalamus modulate spinal ejaculatory autonomic centers.

#### 3.2.3. Spinal Cord-Linked Serotonin

A dense network of 5-HT projections from different brain regions has been demonstrated in rat autonomic and somatic spinal ejaculatory centers [132,213]. There are different 5-HT receptor subtypes in the spinal ejaculatory centers with different cellular localization (pre-/postsynaptic, auto-/heteroreceptors) and different pharmacodynamic properties (ligand affinity, activating/inhibitory). Therefore, it is difficult to understand the exact role of spinal 5-HT in the control of ejaculation [214,215]. From a series of pharmacological studies conducted with different experimental models and yielding apparently contradictory results, it can be concluded that 5-HT exerts a multi-stage and multimodal effect on spinal mechanisms of ejaculation (triggering/facilitating ejaculation, inhibiting expulsion) [134,216,217].

#### 3.2.4. Substance P

Substance P (SP) is a neuropeptide 11amino acids long, the expression of which by the central nervous system (CNS), the peripheral nervous system, and immune cells has been reported in several studies [134,218]. This neuropeptide belongs to a member of the tachykinin (TAC) family of neuropeptides which is encoded by the TAC1 gene. SP exerts its activity through interaction with G protein-coupled neurokinin receptors (NKRs), including NK1R, NK2R and NK3R which are expressed on the surface of many cell types, including neurons [219,220]. SP is one of the neurotransmitters that is known to play a crucial role in the transmission of genital sensory stimuli from the periphery to the spinal cord [221]. Consequently, selective neurotoxic lesion of lumbar spinothalamic neurons, which are characterized by the expression of NK1 receptor, has been reported to negatively influence ejaculation in male rats without significant alteration of other sexual responses [222]. Also, the involvement of the NK1 receptor in the modulation of ejaculation following the intrathecal injection of rats with a selective antagonist has been reported [134,218,219].

The involvement of the NK1 receptor was confirmed in a model of pharmacologically induced ejaculation with intrathecal injection of a selective antagonist [219,223].

#### 3.2.5. Other Neurotransmitters

Neurochemical characterization of the spinal ejaculatory network in the male rat led to the identification of additional factors, including cholecystokinin octapeptide fragment, galanin, gastrin-releasing peptide, and the endogenous opioid peptide enkephalin [134,165]. However, functional evidence is scarce, and the respective role of these factors is poorly understood. Of interest is a pharmacological study showing the pro-ejaculatory action of a gastrin-releasing peptide receptor agonist in the rat [125,134].

## 4. Brain-Linked Transmitters

This group of neurotransmitters includes serotonin, dopamine, and opioids.

### 4.1. Brain-Linked Serotonin

The role of cerebral 5-HT in the control of the ejaculatory response has received particular attention [224,225]. Selective serotonin (5-HT) reuptake inhibitors are widely prescribed as antidepressants and have long been associated with sexual side effects like premature ejaculation [226,227]. A new therapeutic approach for treatment of premature ejaculation was developed and eventually led to the registration of the first agent (dapoxetine) for this indication [228,229,230]. The blockade of 5-HT transporters by SSRIs results in increased 5-HT extracellular levels and, consequently, in enhanced 5-HTergic neurotransmission in the CNS [224,231]. This increased 5-HT tone is suggested to be the mechanism by which SSRIs lengthen ejaculation latency. Whether chronic exposure to selective serotonin (5-H) reuptake inhibitors is more effective than acute exposure in strengthening 5-HT transmission is still controversial [215,232]. It is noteworthy that dapoxetine prolonged ejaculation latency in men diagnosed with premature ejaculation when taken prior to sexual intercourse [228,233]. The clinical results with on-demand dapoxetine have been corroborated in a behavioral rat model of premature ejaculation. Special attention has been paid to the role of cerebral 5-HT in the control of the ejaculatory response [224,225]. Selective serotonin (5-HT) inhibitors, which are administered to patients as antidepressants, have been reported to be associated with the induction of sexual side effects, including premature ejaculation [226,227]. The development of a therapeutic approach for premature ejaculation was successful and led to the approval of the first drug (dapoxetine) for this indication [228,229,230]. It has been reported that the blockade of 5-HT transporters by selective serotonin (5-HT) inhibitors leads to a significant increase in the extracellular 5-HT levels in addition to enhancing the 5-HTergic neurotransmission in the CNS [224,232]. Therefore, increased 5-HT tone is thought to be the mechanism by which selective serotonin reuptake inhibitors prolong ejaculation latency. Whether continuous serotonin exposure in men is more effective than transient exposure and leads to an increased 5-HT transmission remains to be discussed in detail [215,232]. It is noteworthy that the administration of dapoxetine to men suffering premature ejaculation prior to sexual intercourse is expected to prolong ejaculation latency [228,233]. Furthermore, the clinical benefit of dapoxetine administration in men with premature ejaculation has been approved for its clinical validity in a behavioral model of premature ejaculation in rats [228,234].

The mode of action of brain 5-HT in the control of ejaculation has been extensively studied in laboratory animals. An opposite action of 5-HT on sexual behavior, and more particularly ejaculatory behavior, was reported in male rats depending on the site where 5-HT was microinjected [134,198]. While administration of 5-HT to 5-HTergic projection areas (e.g., forebrain, medial preoptic area) has been reported to delay or block ejaculation, administration of 5-HT to brain sites containing 5-HTergic neuronal cell bodies (raphe nuclei) has been found to facilitate ejaculatory behavior in rats [134,235]. The role of 5-HT in the lateral hypothalamus and its effect on ejaculation were functionally analyzed in male rats [235,236]. While higher extracellular 5-HT levels were detected in brain sites with 5-HTergic neuronal cell bodies during the onset of ejaculation, increasing extracellular 5-HT levels by local injection of selective serotonin (5-HT) reuptake inhibitors inhibited sexual behavior, suggesting a central role of 5-HT in regulating the lateral hypothalamus to determine the postejaculatory refractory period [236,237]. The role of 5-HT in the regulation of ejaculation seems to be complicated, besides being mediated by a variety of receptor subtypes that are expressed in the CNS. In addition, the ligands of these receptors, including 5-HT1A, 5-HT1B and 5-HT2C receptors, have been functionally analyzed in various animal models [238,239]. Stimulation of 5-HT1A receptors, known as somatodendritic autoreceptors, has been reported to enhance ejaculatory behavior in male rats [240]. This receptor is expressed on the cell bodies of 5-HT neurons, so that once its activation is switched off, the firing of 5-HT neurons decreases, and consequently 5-HT release is reduced [178]. In addition to the inhibition of ejaculatory behavior in male rats by the stimulation of 5-HT1B receptors, the expression of these receptors has been detected in the medial preoptic area and the lateral hypothalamus [240]. It has been reported that the limbic areas of the nucleus accumbens and amygdala are characterized by increased expression of 5-HT2A/C receptors and are therefore considered to be the site of action of 5-HT2A/C ligands [241,242].

### 4.2. Opioids

Neuroanatomical and functional studies have been conducted to determine the role of other neurochemical factors involved in the cerebral control of ejaculation [134,136], particularly on the inhibitory effects of endogenous opioids and agonists of opioid receptors (mainly l-subtypes) in male rats [134,243].

Opioid receptors are expressed both in the central and peripheral nervous system where they play a pivotal role in analgesia [244,245]. Opioids can be classified according to their effect on opioid receptors. Accordingly, opioids can be considered as agonists, partial agonists and antagonists [244,245]. Agonists interact with opioid receptors to produce a maximal response leading to analgesia [246,247]. Although binding antagonists to opioid receptors does not trigger functional response, antagonists can prevent an agonist from binding to opioid receptors, a discovery which led to the drug naloxone [248,249].

Opioid receptors, as members of the G protein-coupled receptor (GPCR) family [250,251], are subject to agonist-induced internalization [252,253,254], which is associated with desensitization phenomena observed after acute and chronic agonist exposure [255,256]. Most opioid receptor internalization studies have been conducted in cellular models or in vitro studies in which receptor stimulation was produced by agonist administration, thus limiting their physiological relevance [257,258].

The release of endogenous opioids is clearly associated with ejaculation, as demonstrated by the reduced nociception and conditioned place preference that can be elicited by ejaculation in male rats; both phenomena are reversed by the opioid antagonist naloxone [259,260,261]. Furthermore, animals that ejaculate once or copulate to satiation exhibit increased concentrations of leu-enkephalin, met-enkephalin, and octapeptide in the midbrain, cortex, and hypothalamus of male rats [262,263]. The repeated ejaculation in rats might serve as a model of continuous opioid receptor stimulation, provided by a natural stimulus that could be useful to study the dynamics and biological consequences of endogenous opioid-induced receptor internalization in vivo [264].

Accordingly, it is expected that the endogenous opioids released by repeated ejaculation during copulation to satiation might provide continued activation of MOR and delta opioid receptor (DOR) at the ventral tegmental area (VTA), leading to their internalization [257].

Among the non-classical receptors, the opioid growth factor receptor is of particular interest from an oncological perspective [265,266]. Opioid growth factor receptor does not belong to the family of GPCR but rather is a nuclear-associated receptor that is distinct in homology compared to the mu opioid receptor (MOP), kappa opioid receptor (KOP) or delta opioid receptor (DOP) [265,266].

In contrast to the proliferative effect of MOP, the role of KOP in angiogenesis and neovascularization may be inhibitory [265,266]. A study on pluripotent stem cell lines demonstrated the inhibitory effect of KOP agonist on endothelial cell differentiation and vascular formation, as evidenced by the inhibition of vascular endothelial growth factor (VEGF) [265,267].

Although the effects varied between cell lines and ligands as evidenced in vitro studies [184,185,268,269], opioid receptor agonism has been reported to inhibit tumor cell proliferation including PCa [270,271].

### 4.3. Dopamine

Another neurotransmission system that has been extensively studied is the dopaminergic system, and particularly the incertohypothalamic pathway [134,272]. The dopaminergic incertohypothalamic pathway consists of neurons, the cell bodies of which lie in the medial zona incerta and the periventricular nucleus and the terminals of which are in the hypothalamus (including lateral and medial preoptic areas and parvocellular division of the paraventricular nucleus), the thalamus, and the central gray area of the midbrain [273,274]. Dopamine has pro-ejaculatory activity, and the medial preoptic area is a key component of the incertohypothalamic pathway in mediating this effect [275,276]. It was initially unclear which dopamine receptor subtypes were involved because selective pharmacological ligands were not readily available. The dopamine D2-like receptors, which include subtypes D2, D3, and D4, have been found to be involved, but only recently was the crucial role of D3 receptors revealed in male rats by using highly selective D3 antagonists [277,278]. Treatment of schizophrenia in patients with premature ejaculation with antipsychotics (e.g., antagonists of D2-like receptors) or antipsychotic therapeutics (e.g., levosulpiride) has resulted in prolonged ejaculatory latency [134,279].

### 4.4. Brain-Linked Oxytocin

As is known from spinal pharmacology, the oxytocinergic parvocellular neurons in the paraventricular nucleus of the hypothalamus project to several areas of the central nervous system, including brain structures belonging to brain networks associated with ejaculation [134,165]. Brain oxytocin has been shown to play an important role in controlling ejaculation in male rats. Stimulation and blockade of oxytocin receptors in the brain facilitate and inhibit ejaculatory behavior, respectively [134,137]. Thus, a direct effect on the oxytocin pathway and modulation of 5-HT and dopamine neurotransmission likely involves mechanisms that mediate the effects of oxytocin on the ejaculatory response [280,281]. In humans, clinical studies demonstrated no significant effect of oxytocin antagonists on ejaculation time in patients with premature ejaculation [225,282].

### 4.5. Brain-Linked Other Neurotransmitters

Behavioral and neuroanatomical data support the role of pro-opiomelanocortin peptides (adrenocorticotropin and C-melanocyte-stimulating hormone) in the cerebral control of ejaculation in male rats [283,284]. Beneficial effects of these neuropeptides are mediated through the activation of high-affinity receptors, particularly in the medial preoptic area and the lateral hypothalamus. Immunohistochemical studies have identified several neuropeptides in the ejaculatory circuits of the rat brain [134,285]. Among them, cholecystokinin (octapeptide fragment), galanin, and tuberoinfundibular peptide have been found in central brain structures, but their respective roles in controlling the ejaculation process remain to be elucidated [134,285].

## 5. Hormonal Control of Orgasm and Ejaculation

The regulation of all aspects of male reproduction from sperm production to sex drive is a hormone-dependent mechanism [10,286]. Evidence derived from both animal models and human clinical studies implicates the involvement of several hormones that control the ejaculatory process [230,287]. Both ejaculation and orgasm are the consequence of the interplay between the central and the peripheral nervous systems via mechanisms mediated by several neurotransmitters such as dopamine, norepinephrine, serotonin, acetylcholine, gamma-aminobutyric acid (GABA), and nitric oxide [288,289]. Of note, oxytocin, prolactin, thyroid hormones, glucocorticoids, and sex steroid hormones play an active role in the regulation of the process of ejaculation and orgasm [290,291].

Ejaculation is mainly controlled by the autonomic nervous system. It consists of two main phases: emission and ejection [132,137,145]. The main organs involved in ejaculation are the distal epididymis, the vas deferens, the seminal vesicle, the prostate, the prostatic urethra, and the bladder neck [132,145].

The emission phase is a multistep process that begins first with the closure of the bladder neck to prevent retrograde spillage of seminal fluid into the bladder [292,293]. Subsequently, the prostatic secretion, which contains acid phosphatase, citric acid and zinc and is mixed with sperm, is expelled from the vas deferens into the urethra of the prostate [145,294]. Subsequently, the next step is the alkalinization of the final ejaculatory fluid by fructose-containing seminal vesicle fluid [137,295]. The seminal vesicle produces most of the components of the final semen; the Cowper’s glands and the periurethral glands only contribute a small fraction of the semen [296,297]. Both the Cowper’s and periurethral glands have been reported to receive a dense autonomic nerve supply, mainly from the sympathetic and parasympathetic nervous systems originating from the pelvic plexus [132,137]. While the pelvic plexus lies retroperitoneally on the side of the rectum and lateral and posterior to the seminal vesicle, it was found to receive neural input from both the hypogastric and pelvic nerves as well as from the caudal paravertebral sympathetic chain. Accordingly, it is assumed that sympathetic neurons play the predominant role in the ejaculation process, since the nerve endings of sympathetic neurons are characterized by their ability to secrete primarily norepinephrine [298]. Although other neurotransmitters, such as acetylcholine and non-adrenergic/non-cholinergics, have been reported to play an important role in the regulation of the emission phase [137,299], the role of the hypogastric plexus in the regulation of the emission is the most clearly clinically demonstrated [300]. Emission loss was observed after the dissection of non-nerve-sparing para-aortic lymph nodes in testicular cancer [301]. Emission has been induced in paraplegic men through electrical stimulation of the superior hypogastric plexus [137,302]. Also, the input from genital stimulation is integrated at the neural sacral spinal level to produce emission [133,289]. Although the emission phase of ejaculation is under cerebral control, its induction can be targeted to physical or visual erotic stimulation [225,303].

### 5.1. Brain-Linked Oxytocin

As the first phase of the ejaculation process, the ejaculation phase is associated with the release of oxytocin, which is responsible for positive things such as attraction and sexual desire [137,145]. In addition to estrogen, oxytocin is one of the most common hormones involved in the regulation of the ejaculation process [287,304].

Oxytocin is part of a large family of neurohypophysial peptides that share a high degree of homology [158,305]. Oxytocin can bind to oxytocin receptors, a family of nonapeptide hormone receptors including the vasopressin receptors [306,307].

The initiation of oxytocin-dependent mechanisms requires the binding of the oxytocin ligand to the anoxytocin receptor, leading to the activation of signaling pathways which determine the outcome of oxytocin receptor activation [164,308]. The most common oxytocin-stimulated signaling pathways lead to cell proliferation, growth, inhibition, and contraction [164,309]. The binding of oxytocin to the internalized oxytocin receptor activates β-type phospholipase C (PLC-β) receptor via Gαq/11, the subunit of G-coupled protein receptor, which activates PLCβ leading to the hydrolysis of phosphatidylinositol 4,5-bisphosphate (PIP2) to inositol 1,4,5-triphosphate and diacylglycerol (DAG) to activate protein kinase C (PKC) which, in turn, activates mitogen-activated protein kinase (MAPK) and subsequently triggers transient phosphorylation of extracellular regulated kinase 1/2 (ERK1/2), leading to proliferation [310,311].

External stimulation of the oxytocin receptor by its ligand triggers three different pathways via Gαi-dependent mechanisms that lead to the enhancement of the antiproliferative effect and the contraction of smooth muscle [281,312]. Thus, the antiproliferative effect induced by oxytocin is mediated by PLCβγ, leading to sustained phosphorylation of ERK1/2 and subsequent expression of the cyclin-dependent kinase inhibitor p21-dependent signaling pathway, and by cyclic adenosine monophosphate (cAMP), leading to activation of the protein kinase C (PKC)-dependent pathway [313,314,315]. Meanwhile, oxytocin-induced smooth muscle contraction is mediated by the activation of the PLC signaling pathway, which leads to the release of PIP2 to IP3 and DAG to activate PKC, and by increasing calcium release from the sarcoplasmic reticulum, which activates the calmodulin complex, which enhances the activation of MLC, leading to contraction [315].

In addition, the DAG pathway can activate protein kinase C to enhance MAPK pathway activation and ultimately the production of prostaglandin [281,316]. Oxytocin-induced activation of cytosolic phospholipase A2 (cPLA2) increases the production of prostaglandin, leading to contraction [317,318]. Oxytocin-induced activation of the Rho/Rho-associated protein kinase (ROCK) pathway increases the phosphorylation of MLC1 and subsequently contraction enhancement [319,320].

Taken together, the increased levels of oxytocin produced in response to frequent ejaculation is implicated in the prevention of transformation of normal prostate cells into tumor cells [321,322], a contributing role for elevated oxytocin in PCa risk reduction. The mechanisms of oxytocin-induced pathways in the prostate are outlined in Figure 2.

### 5.2. Estrogens

The primary hormonal functions of estrogen are mediated by estrogen-specific receptors, of which there are several types with potentially opposing functions. These different receptors are distributed in various human tissues at different expression levels [322,323]. Importantly, several ERs are localized in prostate tissue and their activation has been associated with several phenotypic changes in vitro and in vivo. There is considerable evidence that direct estrogen signaling in prostate cells plays an important role in the development of the prostate gland and possibly also in the development of cancer [22,324]. Most of the hormone-regulating effects of estrogens are attributed to nuclear steroid alpha (ERα) and estrogen receptor β (ER β) [325]. Accumulated evidence demonstrates that rapid membrane-initiated signaling of estrogens can also occur through the stimulation of estrogen receptor α or β located in the cell membrane, leading to the initiation of downstream kinase cascades [326,327].

The most common pleiotropic effects of these hormones in males are associated with the development of the reproductive tract’s structure, and therefore fertility [328,329]. In addition to their physiological role in controlling testosterone levels and bone mineral homeostasis, both natural estrogen and its analogs have been reported to prevent the development of PCa, which is evidence for the involvement of estrogen in the reduction of tumor risk [330,331].

Apart from their regulatory function in the male reproductive system, both Erα and ER β are implicated in the growth and differentiation of the prostate gland under normal and pathological conditions [9,331].

Functional analysis of the anticancer properties of estrogen analogs, such as genistein derived from soy isoflavones, has been reported [332,333]. Phytoestrogen, an estrogen-like compound derived from plants, has been reported for its impact on PCa risk reduction [334,335]. Also, the analysis of the products of both fermented and nonfermented soy-derived estrogens exhibited protective effects against PCa [336,337]. The prevention of carcinogenesis in the dorsolateral and anterior prostate lobes by the treatment of estrogen homologs such as genistein and diazein suggests an important role for frequent ejaculation-associated estrogen in PCa risk reduction [338,339].

As a member of the nuclear receptor family, ERβ acts individually to form homodimers (ERβ/β) or heterodimers (ERβ/α). Thus, ligand-induced dimerization of ERβ is essential to facilitate the translocation of ERβ dimers to the nucleus, where they can bind with co-regulatory proteins and interact with nuclear factor-κB (NF-κB) binding sites and activator protein 1 (AP-1) in promoter regions of different target genes, largely characterized by their anti-proliferative effects [340,341]. Consequently, frequent ejaculation-induced production of estrogen is essential to prevent PCa development and to reduce tumor risk [340,341]. A proposed model for the mechanisms of estrogen-induced antiproliferative effects in prostate tissues is outlined in Figure 3.

Although both ERα and ERβ are similar in structure, they exert different effects on oncogenes [342,343]. While Erβ exhibits a suppressive effect, Erα has been shown to have oncogenic potential [344,345].

The abundant localization of both ERα and ERβ in the epididymal cauda is essential in the regulation of epididymal function [346,347]. Thus, in the ejaculation phase, estrogen influences the contractility of the epididymis and thereby affects latency time [348,349].

In the adult human prostate, ERβ is an important mediator of epithelial differentiation [341,350]. The mechanisms through which ERβ maintains epithelial differentiation have been reported and are thought to be regulated by the degradation of hypoxia-inducible factor 1α (HIF-1α) [351]. The translocation of the ERβ dimer to the nucleus in response to the stimulation by its ligand is mostly associated with the enhancement of the transcription of prolylhydroxylase domain-containing protein 2 (PHD2), which hydroxylates HIF-1α via a von Hippel–Lindau tumor suppressor (VHL)-dependent mechanism [341,351]. However, the antiproliferative actions of ERβ are independent from alteration of systemic androgen concentrations and the activation of Erα [352,353].

## 6. Significance of Frequent Ejaculation in the Reduction of Prostate Cancer Risk

The mechanisms regulating frequent ejaculation-associated reduction of PCa risk are attributed to the suppression of the sympathetic nervous system, leading to relief of tension and the slowing of prostate epithelial cell division [310,339]. Of note, if the ejaculation frequency impacts gene expression in the prostate tissue, then gene expression can influence tissue susceptibility to tumorigenesis and provide insight into the mechanisms whereby ejaculation affects PCa risk.

Due to the difficulty in establishing a connection between frequent ejaculation and gene expression patterns of prostate tissues in the context of health status and other lifestyle factors [354,355,356], more research is needed to uncover why frequent ejaculation may protect men from PCa. While a differential gene expression analysis in prostate tissue in the context of frequent ejaculation demonstrated some limitations [357,358], the analysis of differential gene expression profile and pathways in prostate tumor areas and in the adjacent normal tissues in the context of frequent ejaculation revealed that the identified genes and pathways serve as potential biological links between the frequency of ejaculation and prostate tumorigenesis [355]. Thus, the investigation of the biological basis of frequent ejaculations leading to the reduction of PCa risk may help develop a management protocol for the prevention or the reduction of PCa occurrence in men [354,355].

In addition to the prostate stagnation hypothesis [359,360], several mechanisms have been proposed to explain an inverse association between ejaculation frequency and PCa. The greater the frequency of ejaculation, the more the function of epithelial cells in the peripheral zone is impaired, thereby hindering the metabolic switch from citrate secretion to citrate oxidation, thus leading to the delay of tumor initiation in the prostate [354,358]. Alternatively, frequent ejaculation may reduce the development of intraluminal crystalloids in the prostate, which are associated with a higher risk of PCa tumorigenesis [354,355]. A higher ejaculation frequency may be associated with a reduction in psychological tension and a suppression of the sympathetic central nervous system, which could attenuate prostate epithelial cell division [360]. Since no modifiable risk factors for PCa have been identified to date, the specific biological mechanisms underlying these associations deserve further investigation.

## 7. Crosstalk Between Androgen Receptors and Endothelial Growth Factor Signaling Pathways in the Prostate

Remarkably, a drastic change in three hormones, prolactin, cortisol, and testosterone, occurs post ejaculation [361,362]. While androgen hormone/testosterone levels increase dramatically via ejaculation, any changes to prolactin and cortisol levels occur before and after ejaculation [361,362].

Androgens modulate prostate growth and function through a multi-step mechanism [36,363]. This multi-step mechanism is mediated by the conversion of testosterone to 5α-DHT via the enzyme 5α-reductase [363,364]. Consequently, 5α-DHT binds to AR and forms a 5α-DHT:AR complex which undergoes molecular changes called “activation and transformation”, leading to specific interactions between the 5α-DHT:AR complex and a specific DNA enhancer element, as well as androgen response elements specific to androgen-regulated genes [31,365]. The results are an upregulation of androgen-specific target genes and a downregulation of other genes to maintain homeostasis. Specific target genes downregulated by androgens in normal prostate tissue include the endothelial growth factor receptor (EGFR) [36,366].

The induction of EGFR expression by androgens is regulated at the transcriptional level, both in normal prostate tissue and prostate cancer tissue [367,368]. However, EGFR expressions seem to be down-regulated in normal prostate tissues and upregulated in PCa cells, particularly androgen-independent PCa [367,368]. Castration of mature animals demonstrated time-dependent increases in EGFR protein expression in the prostate, whereas the treatment of castrated animals with testosterone or 5α-DHT reduced EGFR protein expression [36,368], suggesting that expression and maintenance of EGFR in normal prostate tissue occurs through androgen-dependent inhibition. Therefore, frequent ejaculation-associated production of androgens is expected to reduce PCa risk through the inhibition of EGFR expression. A proposed model for androgens/testosterone-mediated attenuation of EGFR in normal prostate tissues is outlined in Figure 4.

## 8. Mechanisms of Frequent Ejaculation-Induced Reduction of Prostate Cancer Risk

The regulation of PCa risk induced by frequent ejaculation appears to be mediated by several mechanisms that have been proposed to explain the association between frequent ejaculation and reduced PCa [369,370]. The higher the frequency of ejaculation, the more inhibited the metabolic switch from citrate secretion to citrate oxidation in the epithelial cells in the peripheral zone. This is one of the possible mechanisms by which frequent ejaculation may reduce the risk of tumor development in the prostate [370]. Also, frequent ejaculation is expected to reduce PCa risk via mechanisms mediated by restoration of the development of intraluminal eosinophilic structures in the prostate, which are considered a sign of higher risk for PCa [371,372]. Another mechanism by which frequent ejaculation can reduce PCA risk is expected to be mediated by reducing psychological tension and suppressing the central sympathetic nervous system, resulting in a significant attenuation of the stimulation of prostate epithelial cell division [373].

As is generally accepted, cancer is one of the most common diseases caused by environmental and external influences [374,375]. The development of around 90% of cancers diagnosed worldwide is attributed mainly to harmful environmental chemicals [376]. Among these substances, several active harmful chemicals are recognized as endocrine disruptors [377,378]. These endocrine disruptors have common characteristics that are mostly associated with disruption of hormone regulation and action [379]. Endocrine disruptors act primarily via nuclear hormone receptors, such as estrogen and progesterone receptors [380,381]. The mechanisms of endocrine disruptors are much more comprehensive than commonly reported [382,383].

Like other organs, the prostate is a target of numerous environmentally harmful chemicals, including endocrine disruptors, which can induce multi-organ disease through mechanisms based on interaction with the cellular receptors or components [384,385]. Therefore, the interference of these harmful chemicals with the action of hormones, which are involved in the different cellular functions of the prostate gland, is expected to provoke hormone-dependent malignancies [386,387]. Ultimately, many reports have suggested that endocrine disruptors have the potential to significantly alter the epigenomic landscape in cancers, including prostate cancer [388,389]. Thus, frequent ejaculation is expected to protect the prostate by flushing out these harmful chemicals. Consequently, men who ejaculate more frequently can reduce their biological load of harmful chemicals, and that can, in turn, reduce the risk of prostate cancer.

The stimulation of endocannabinoid release in humans by masturbation to orgasm suggests an essential role for frequent ejaculation in the reduction of prostate cancer via the endogenous 2-arachidonoylglycerol (2-AG)-dependent mechanism [390]. Also, several studies have demonstrated the ability of 2-AG to inhibit the invasion of androgen-independent prostate cancer PC-3 and DU-145 cell lines in vitro [390].

## 9. Conclusions

Burgeoning interest in the relationship between ejaculation frequency and prostate cancer risk has grown in recent years. Increased ejaculation without risky sexual behavior could be an important means of reducing the significant medical costs and physical and psychological side effects of unnecessary diagnosis and treatment of low-risk tumors, even though these appear to be less strongly associated with the aggressive form of the disease. Although accumulated evidence indicates a significant role for ejaculation frequency in the reduction of PCa risk, the biological mechanisms underlying the reduction in PCa risk mediated by frequent ejaculation have not been explored. Nonetheless, investigating the relationship between ejaculation frequency and prostate cancer risk is of great importance because of its potential implications for public health and prevention strategies. The main aim of this review is to shed light on possible mechanisms involved in regulating ejaculation frequency and PCA risk reduction. Although some crucial gaps remain unexplained, the description of a brain network responsible for controlling the ejaculatory response and the identification of a spinal generator of ejaculation represents an important milestone.

## Figures and Tables

**Figure 1 cancers-17-00843-f001:**
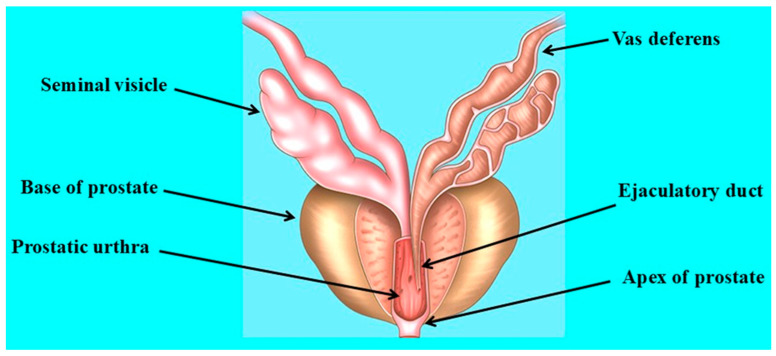
Functional structure of the prostate gland.

**Figure 2 cancers-17-00843-f002:**
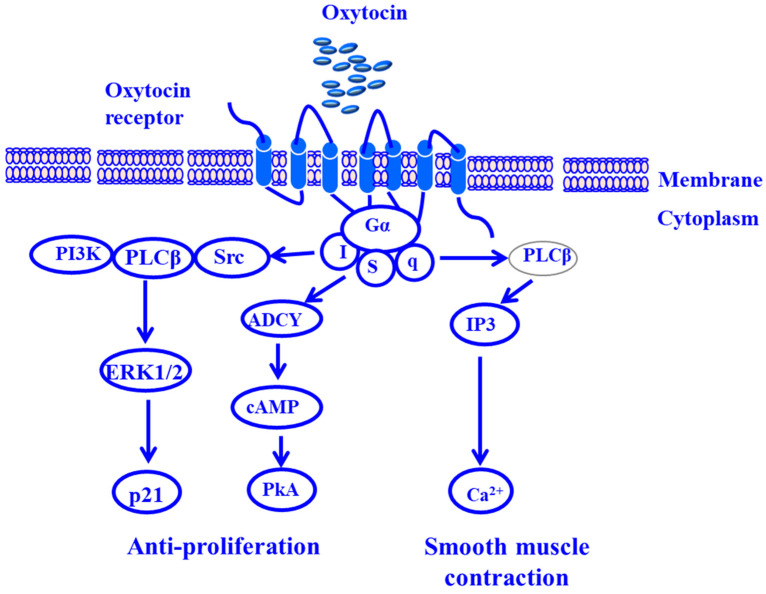
Schematic illustration proposing routes for oxytocin receptor signaling. Upon ligand binding, oxytocin becomes able to couple via proteins of the Gαq/11 family to phospholipase C-b (PLC-b), which catalyzes the hydrolysis of phosphatidylinositol 4,5-bisphosphate (PIP2) to inositol 1,4,5-triphosphate and diacylglycerol (DAG). Inositol 1,4,5-triphosphate will stimulate the release of intracellular calcium stores from the sarcoplasmic reticulum, leading to smooth muscle contraction. The DAG pathway can then activate protein kinase C (PKC), triggering the phosphorylation of the mitogen-activated protein kinase (MAPK) pathway which leads to proliferation. The interaction of the oxytocin receptor with Gα inhibitor (Gα_i_) in the prostate results in sustained phosphorylation of ERK to trigger anti-proliferative effects. Coupling of the oxytocin receptor to Gα_i_ leads to an antiproliferative outcome via protein kinase A (PKA)-dependent mechanisms.

**Figure 3 cancers-17-00843-f003:**
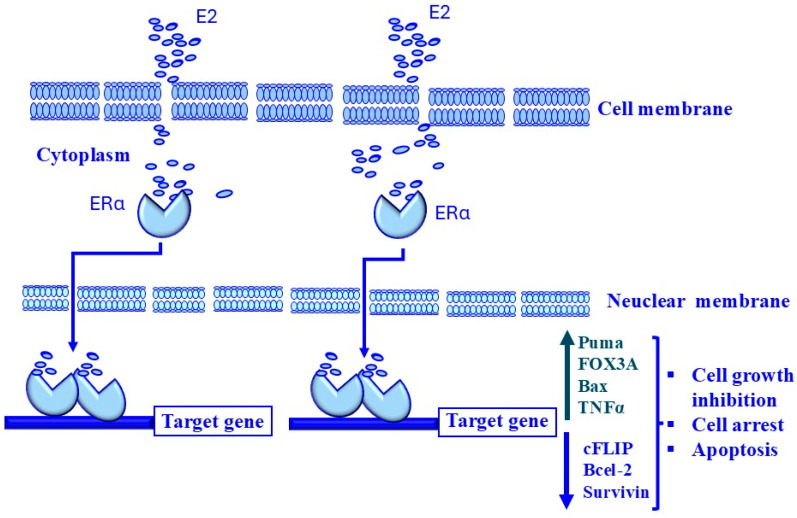
Biological consequences of estrogen receptor α (Erα) activation in the prostate. ERα activation upon estrogen binding (E2) leads to its translocation into the nucleus. There, ERα homodimers bind DNA either directly or indirectly, through estrogen responsive elements (EREs), or upon binding to other transcription factors, which leads to the induction or suppression of the transcriptional activity of genes associated with cell growth inhibition, cell arrest, and apoptosis.

**Figure 4 cancers-17-00843-f004:**
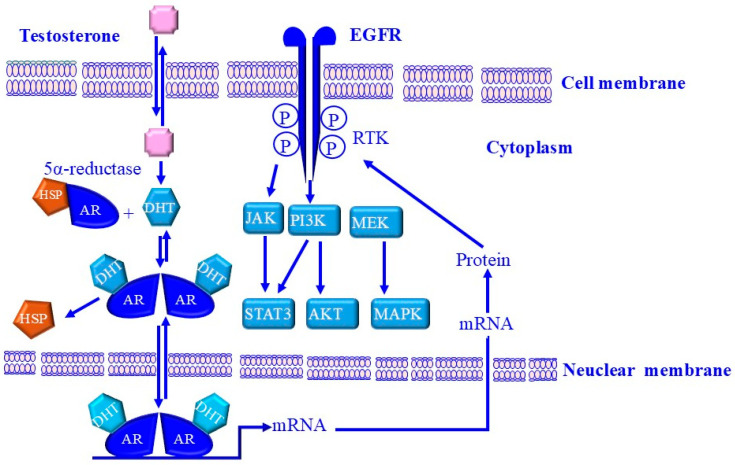
Androgen receptor (AR) signaling in normal and malignant prostate tissue. Testosterone is transported by a simple diffusion into the cell to be metabolized by the 5α-reductase enzyme to produce a more potent androgen, 5α-dihydrotestosterone (5α-DHT). Binding of 5α-DHT to the androgen receptor (AR) leads to AR activation and transformation via a mechanism mediated by its dissociation from heat-shock proteins, dimerization and phosphorylation. Consequently, the activated and transformed 5α-DHT. AR complex becomes able to translocate into the nucleus, where it interacts with the androgen response element. Binding of the 5α-DHT. AR complex to the specific DNA response elements leads to recruitment of co-activators or co-repressors to regulate gene expression. In normal prostate tissue, the activation of AR triggers the downregulation of the epidermal growth factor receptor (EGFR) and subsequently the reduction of the active functional protein. In malignant prostate tissue, the molecular switch is turned off, leading to increased protein synthesis and subsequently the increase of functional EGFR. EGFR activation by EGF in malignant prostate tissue is expected to trigger the activation of AR in the presence and in absence of 5α-DHT. A change in EGFR expression and functional activity leads to androgen independence of the tumor. Loss of androgen regulation associated with increased EGFR expression or signaling in malignant prostate tissue via the PI3K and MAPK pathways leads to androgen-independent tumor growth. These signaling pathways can activate the AR without ligand, thereby enhancing androgen receptor signaling, leading to cell proliferation, migration and survival.

## Data Availability

Not applicable.

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
