# Peer review of "Reduction of Prostate Cancer Risk: Role of Frequent Ejaculation-Associated Mechanisms"

_cancers, 2025, doi:10.3390/cancers17050843_

Round 1

Reviewer 1 Report

Comments and Suggestions for Authors

Mechanisms Regulating the Reduction of Prostate Cancer Risk

According to my understanding from the title, the authors aim to talk about the general mechanisms regulating the reduction of prostate cancer risk. However, upon reading through, it becomes evident that the authors emphasize the effects of frequent ejaculation on reducing prostate cancer risk. More surprisingly, the majority of the content describes factors affecting ejaculation rather than the mechanism by which frequent ejaculation reduces prostate cancer risk. Therefore, I found this review lacks a clear direction regarding its main message. If the authors want to focus on how frequent ejaculation benefits prostate cancer risk reduction, they should change the title accordingly and focus on the mechanism by which frequent ejaculation reduces prostate cancer risk.

Major Issues:

  • As mentioned above, the authors need to determine their primary focus and modify the entire review accordingly, especially the title and overall structure.

Minor Issues:

  1. Some sentences lack periods, such as those on line 80 and line 210. Please review the entire manuscript for similar errors.
  2. Regarding the sentence "The more frequent ejaculation occurs without risky sexual behavior, the more the risk of prostate cancer decreases in men,” consider changing it to "The more frequent ejaculation occurs without risky sexual behavior, the less the risk of prostate cancer."
  3. Several paragraphs are underlined. What does this signify?
  4. There are repetitive sentences throughout the manuscript, such as lines 34-36 and lines 43-44. Please review the entire manuscript for similar issues.
  5. For the description "The prostate gland is part of the male reproductive system located inside the body," could you please modify it to a more professional phrasing, such as the description on lines 54-55?
  6. In the author contribution section, please omit the instructional content.

Author Response

Author response to Reviewer 2

Dear Editor,

Enclosed find please our response to the valuable comment of reviewer 01

Reviewer 1

Comments and Suggestions for Authors

Mechanisms Regulating the Reduction of Prostate Cancer Risk

According to my understanding from the title, the authors aim to talk about the general mechanisms regulating the reduction of prostate cancer risk. However, upon reading through, it becomes evident that the authors emphasize the effects of frequent ejaculation on reducing prostate cancer risk. More surprisingly, the majority of the content describes factors affecting ejaculation rather than the mechanism by which frequent ejaculation reduces prostate cancer risk. Therefore, I found this review lacks a clear direction regarding its main message. If the authors want to focus on how frequent ejaculation benefits prostate cancer risk reduction, they should change the title accordingly and focus on the mechanism by which frequent ejaculation reduces prostate cancer risk.

Major Issues:

Comment 1: As mentioned above, the authors need to determine their primary focus and modify the entire review accordingly, especially the title and overall structure.

Authors’ response: Thank you very much for your valuable comment. Accordingly, we modified the title to be more suitable and we made significant modification and changes in the overall structure of the manuscript. Accordingly, the following modifications and changes have been made.

See lines: 2-3; The current title [Mechanisms Regulating the Reduction of Prostate Cancer Risk ] has been modified and replaced by the following title [Reduction of Prostate Cancer Risk: Role of Frequent Ejaculation-Associated Mechanisms]

See Lines:794-831; the following chapter [8. Mechanisms of frequent ejaculation-induced reduction of prostate cancer risk The regulation of PCa risk induced by frequent ejaculation appears to be mediated by several mechanisms that have been proposed to explain the association between frequent ejaculation and reduced PCa [372, 373]. The higher the frequency of ejaculation, the more inhibited the metabolic switch from citrate secretion to citrate oxidation in the epithelial cells in the peripheral zone. One of the possible mechanisms by which frequent ejaculation may reduce the risk of tumor development in the prostate [373]. Also, frequent ejaculation is expected to reduce PCa risk via mechanism mediated by restore the development of intraluminal eosinophilic structures in the prostate, which are considered a sign of higher risk for PCa. [374, 375]. Another mechanism by which frequent ejaculation can reduce PCA risk is expected to be mediated by reducing psychological tension and suppressing the central sympathetic nervous system, resulting in a significant attenuation of the stimulation of prostate epithelial cell division. [376].

As is generally accepted, cancer is one of the most common diseases caused by environmental and external influences [377, 378]. The development of around 90% of cancers diagnosed worldwide is attributed mainly to harmful environmental chemicals [379]. Among these substances, several active harmful chemicals that are recognized as endocrine disruptors [380, 381]. These endocrine disruptors have common characteristics that are mostly associated with disruption of hormone regulation and action [382]. Endocrine disruptors act primarily via nuclear hormone receptors, such as estrogen and progesterone receptors [383, 384]. The mechanisms of endocrine disruptors are much more comprehensive than commonly reported [385, 386].

Like other organs, the prostate is a target of numerous environmentally harmful chemicals including endocrine disruptors that can induce multi-organ disease through mechanisms based on interaction with the cellular receptors or components [387, 388]. Therefore, the interference of these harmful chemicals with the action of hormones, which are involved in the different cellular functions of the prostate gland is expected to develop hormone-dependent malignancies [389, 390]. Ultimately, many reports suggested that endocrine disruptors have the potential to significantly alter the epigenomic landscape in cancers, including prostate cancer [391, 392]. Thus, frequent ejaculation is expected to protect the prostate by flushing out these harmful chemicals. Consequently, men who ejaculate more frequently can have healthy lifestyle habits harmful chemicals that can, in turn, reduce the risk of prostate cancer.

The stimulation of endocannabinoid release in human by masturbation to orgasm suggests an essential role for the frequent ejaculation in the reduction of prostate cancer via endogenous 2-arachidonoylglycerol (2-AG) -dependent mechanism [393]. Also, several studies demonstrated the ability of 2-AG to inhibit the invasion of androgen-independent prostate cancer PC-3 and DU-145) cell lines in vitro [393]. ] has been added to the main text.

See lines:1447-1802;

Lozano-Lorca, M.; Olmedo-Requena, R.; Barrios-Rodríguez, R.; Jiménez-Pacheco, A.; Vázquez-Alonso, F.; Castillo-Bueno, H. M.; Rodríguez-Barranco, M.; Jiménez-Moleón, J. J. Ejaculation Frequency and Prostate Cancer: CAPLIFE Study. World J Mens Health 202341 (3), 724-733. DOI: 10.5534/wjmh.220216

Costello, L. C.; Franklin, R. B. The clinical relevance of the metabolism of prostate cancer; zinc and tumor suppression: connecting the dots. Mol Cancer 20065, 17. DOI: 10.1186/1476-4598-5-17.

Svatek, R. S.; Karam, J. A.; Rogers, T. E.; Shulman, M. J.; Margulis, V.; Benaim, E. A. Intraluminal crystalloids are highly associated with prostatic adenocarcinoma on concurrent biopsy specimens. Prostate Cancer Prostatic Dis 200710 (3), 279-282. DOI: 10.1038/sj.pcan.4500954.

Del Rosario, A. D.; Bui, H. X.; Abdulla, M.; Ross, J. S. Sulfur-rich prostatic intraluminal crystalloids: a surgical pathologic and electron probe x-ray microanalytic study. Hum Pathol 199324 (11), 1159-1167. DOI: 10.1016/0046-8177(93)90210-8.

Newman, H. F.; Reiss, H.; Northup, J. D. Physical basis of emission, ejaculation, and orgasm in the male. Urology 198219 (4), 341-350. DOI: 10.1016/0090-4295(82)90186-8.

Parsa, N. Environmental factors inducing human cancers. Iran J Public Health 201241 (11), 1-9.

Baillie-Hamilton, P. F. Chemical toxins: a hypothesis to explain the global obesity epidemic. J Altern Complement Med 20028 (2), 185-192. DOI: 10.1089/107555302317371479.

Madia, F.; Worth, A.; Whelan, M.; Corvi, R. Carcinogenicity assessment: Addressing the challenges of cancer and chemicals in the environment. Environ Int 2019128, 417-429. DOI: 10.1016/j.envint.2019.04.067.

Jensen, T. K.; Andersson, A. M.; Jørgensen, N.; Andersen, A. G.; Carlsen, E.; Petersen, J. H.; Skakkebaek, N. E. Body mass index in relation to semen quality and reproductive hormones among 1,558 Danish men. Fertil Steril 200482 (4), 863-870. DOI: 10.1016/j.fertnstert.2004.03.056.

Macdonald, A. A.; Stewart, A. W.; Farquhar, C. M. Body mass index in relation to semen quality and reproductive hormones in New Zealand men: a cross-sectional study in fertility clinics. Hum Reprod 201328 (12), 3178-3187. DOI: 10.1093/humrep/det379.

[296] La Merrill, M. A.; Vandenberg, L. N.; Smith, M. T.; Goodson, W.; Browne, P.; Patisaul, H. B.; Guyton, K. Z.; Kortenkamp, A.; Cogliano, V. J.; Woodruff, T. J.; et al. Consensus on the key characteristics of endocrine-disrupting chemicals as a basis for hazard identification. Nat Rev Endocrinol 202016 (1), 45-57. DOI: 10.1038/s41574-019-0273-8.  

Lee, H. R.; Jeung, E. B.; Cho, M. H.; Kim, T. H.; Leung, P. C.; Choi, K. C. Molecular mechanism(s) of endocrine-disrupting chemicals and their potent oestrogenicity in diverse cells and tissues that express oestrogen receptors. J Cell Mol Med 201317 (1), 1-11. DOI: 10.1111/j.1582-4934.2012.01649.x.

Tabb, M. M.; Blumberg, B. New modes of action for endocrine-disrupting chemicals. Mol Endocrinol 200620 (3), 475-482. DOI: 10.1210/me.2004-0513..

Diamanti-Kandarakis, E.; Bourguignon, J. P.; Giudice, L. C.; Hauser, R.; Prins, G. S.; Soto, A. M.; Zoeller, R. T.; Gore, A. C. Endocrine-disrupting chemicals: an Endocrine Society scientific statement. Endocr Rev 200930 (4), 293-342. DOI: 10.1210/er.2009-0002.

Ankolkar, M.; Balasinor, N. H. Endocrine control of epigenetic mechanisms in male reproduction. Horm Mol Biol Clin Investig 201625 (1), 65-70. DOI: 10.1515/hmbci-2016-0007 

Corti, M.; Lorenzetti, S.; Ubaldi, A.; Zilli, R.; Marcoccia, D. Endocrine Disruptors and Prostate Cancer. Int J Mol Sci 202223 (3). DOI: 10.3390/ijms23031216.

Bleak, T. C.; Calaf, G. M. Breast and prostate glands affected by environmental substances (Review). Oncol Rep 202145 (4). DOI: 10.3892/or.2021.7971

Lacouture, A.; Lafront, C.; Peillex, C.; Pelletier, M.; Audet-Walsh, É. Impacts of endocrine-disrupting chemicals on prostate function and cancer. Environ Res 2022204 (Pt B), 112085. DOI: 10.1016/j.envres.2021.112085.

Prins, G. S. Endocrine disruptors and prostate cancer risk. Endocr Relat Cancer 200815 (3), 649-656. DOI: 10.1677/ERC-08-0043.

Lehle, J. D.; McCarrey, J. R. Differential susceptibility to endocrine disruptor-induced epimutagenesis. Environ Epigenet 20206 (1), dvaa016. DOI: 10.1093/eep/dvaa016.

Lang, I. A.; Galloway, T. S.; Scarlett, A.; Henley, W. E.; Depledge, M.; Wallace, R. B.; Melzer, D. Association of urinary bisphenol A concentration with medical disorders and laboratory abnormalities in adults. JAMA 2008300 (11), 1303-1310. DOI: 10.1001/jama.300.11.1303. 31, 41, 46].

Fuss, J.; Bindila, L.; Wiedemann, K.; Auer, M. K.; Briken, P.; Biedermann, S. V. Masturbation to Orgasm Stimulates the Release of the Endocannabinoid 2-Arachidonoylglycerol in Humans. J Sex Med 201714 (11), 1372-1379. DOI: 10.1016/j.jsxm.2017.09.016.

Nithipatikom, K.; Endsley, M. P.; Isbell, M. A.; Falck, J. R.; Iwamoto, Y.; Hillard, C. J.; Campbell, W. B. 2-arachidonoylglycerol: a novel inhibitor of androgen-independent prostate cancer cell invasion. Cancer Res 200464 (24), 8826-8830. DOI: 10.1158/0008-5472.CAN-04-3136. have been added to reference section

Minor Issues:

Comment1: Some sentences lack periods, such as those on line 80 and line 210. Please review the entire manuscript for similar errors.

Authors ‘response: Thank you very much for your valuable comment. Accordingly, we corrected the lack of periods overall in the manuscript. 

Comment 2: Regarding the sentence "The more frequent ejaculation occurs without risky sexual behavior, the more the risk of prostate cancer decreases in men,” consider changing it to "The more frequent ejaculation occurs without risky sexual behavior, the less the risk of prostate cancer."

Authors’response: Thank you very much for your comment. Accordingly, we considered your suggestion replaced the sentence as required

See lines:27-28; the following sentence [The more frequent ejaculation occurs without risky sexual behavior, the more the risk of prostate cancer decreases in men] is replaced by the following sentence [The more frequent ejaculation occurs without risky sexual behavior, the less the risk of prostate cancer  ]

Comment 3: Several paragraphs are underlined. What does this signify?

Authors’ response: Thank you very much for your comment. The underlined paragraphs were the authors ‘response to the comment of the editor. Accordingly, we underlined the changes that have been made in the manuscript.

Comment 4: There are repetitive sentences throughout the manuscript, such as lines 34-36 and lines 43-44. Please review the entire manuscript for similar issues.

Authors’ response: Thank you very much for your valuable comment. Accordingly, we made the required correction overall in the manuscript.

See lines:35-36;43-45

Comment 5: For the description "The prostate gland is part of the male reproductive system located inside the body," could you please modify it to a more professional phrasing, such as the description on lines 54-55?

Authors’response: Thank you very much for your comment. As required, we modified the sentence to be more professional.

See lines:37-38; the following sentence [The prostate gland is part of the male reproductive system located inside the body. ] has been modified and replaced by the following sentence [The prostate is a small gland that is anatomically located below the bladder and in front of the rectum]

Comment 6: In the author contribution section, please omit the instructional content.

Authors’response: Thank you very much for your comment. As requested, we omitted the instructural content.

Reviewer 2 Report

Comments and Suggestions for Authors

This is an interesting paper dealing with the relationship between ejaculation frequency and prostate cancer risk and a well-done survey of different groups of neurotransmittors is given.My comments are the following:

page 10/30, third paragraph: The description of the prostatic secretion is too potted. Bivalent cations Mg2+ and Ca2+ should be mentioned together with Zn2+ and most importantly that the prostatic secretion also contains an abundance of subcellular, exosome-like extracellular vesicles denoted "prostasomes". Prostasomes play a crucial role in development of prostate cancer (Prostate Cancer and Prostatic Diseases 7 (2004) 21-31). Moreover, activation of the renin-angiotensin system (RAS) plays an important role in normal physiology. Angiotensin-converting enzyme (ACE) degrades vasodilator kinins and generates angiotensin II, being the major effector peptide in the RAS-system. ACE synthesis by the prostate gland has been reported (Int J Androl 12 (1989) 22-28; J Pathol 195 (2001) 571-579) and the activity of ACE is much higher in seminal plasma than in blood serum and most activity is membrane-bound to prostasomes. Angiotensin II stimulates neovascularization which is one of the requirements for tumor growth. Angiotensin II can also act as a tumor growth factor with stimulation of cell replication and is associated with genes that control cell growth. A cohort study with nested case-control analysis showed a lower risk of subsequent prostate cancer in hypertensive patients medicated with a well-established inhibitor of ACE (The Prostate 58(2004) 50-56). Similar results with reduced prostate cancer risk have been subseqently achieved with hypertensive patients, now medicated with antagonists towards the angiotensin II receptor. This aspect of reduction of prostate risk should be mentioned and briefly discussed by the authors. Minor comment: page 20/30: Reference 81 is incomplete 

Author Response

Author response to Reviewer 2

Dear Editor,

Enclosed find please our response to the valuable comment of reviewer 02

Comments and Suggestions for Authors

This is an interesting paper dealing with the relationship between ejaculation frequency and prostate cancer risk and a well-done survey of different groups of neurotransmittors is given.My comments are the following:

Comment 1: page 10/30, third paragraph: The description of the prostatic secretion is too potted. Bivalent cations Mg2+ and Ca2+ should be mentioned together with Zn2+ and most importantly that the prostatic secretion also contains an abundance of subcellular, exosome-like extracellular vesicles denoted "prostasomes". Prostasomes play a crucial role in development of prostate cancer (Prostate Cancer and Prostatic Diseases 7 (2004) 21-31).  Moreover, activation of the renin-angiotensin system (RAS) plays an important role in normal physiology. Angiotensin-converting enzyme (ACE) degrades vasodilator kinins and generates angiotensin II, being the major effector peptide in the RAS-system. ACE synthesis by the prostate gland has been reported (Int J Androl 12 (1989) 22-28; J Pathol 195 (2001) 571-579) and the activity of ACE is much higher in seminal plasma than in blood serum and most activity is membrane-bound to prostasomes. Angiotensin II stimulates neovascularization which is one of the requirements for tumor growth. Angiotensin II can also act as a tumor growth factor with stimulation of cell replication and is associated with genes that control cell growth. A cohort study with nested case-control analysis showed a lower risk of subsequent prostate cancer in hypertensive patients medicated with a well-established inhibitor of ACE (The Prostate 58(2004) 50-56). Similar results with reduced prostate cancer risk have been subseqently achieved with hypertensive patients, now medicated with antagonists towards the angiotensin II receptor. This aspect of reduction of prostate risk should be mentioned and briefly discussed by the authors.

Authors’response: Thank you very much for your valuable comment. Accordingly,  we added several paragraphs  as you suggested. The following changes in the manuscript have been made. The title of chapter 2  to be more suitable

See line: 101   The following Chapter title [2. Anatomy and Physiological Functions of Prostate ] has been replaced by the following  one [2. Anatomy, Physiological Functions and secretion of Prostate ].  

See lines:116-247; the following text [As an additional sexual organ, the prostate secretes fluid into the semen, which supports sperm release and fertility [8, 46]. The cells of the prostate can be roughly divided into two cell types: the parenchymal epithelial cells that form the acini and ducts of the gland and secrete prostatic fluid, and the stromal cells, which surround the acini and ducts and provide support and structure to the gland [46, 47].

The main function of the accessory sex tissues, which include the prostate, seminal vesicles and urethral glands, is the production of seminal plasma, the non-gametic portion of semen [48, 49]. The main function of the seminal plasma is to serve as a buffered, nutrient-rich transport medium for sperm when they are deposited in the female genital tract [50, 51]. Seminal plasma is characterized by its alkalinity that is essential to neutralize the acidic environment of the vagina [52, 53]. While the main function of the prostaglandins hat are present in seminal plasma is to stimulate the contraction of smooth muscles along the genital tract of the female to facilitate the transport of the sperm to the egg [54,55]. In addition to zinc and IgA that are characterized by their bacteriostatic properties, the seminal plasm many prostatic proteins that are essential to prevent autoagglutination of sperm [56, 57].

As is known, in normal men, seminal plasma production, including the prostatic component, is regulated by circulating androgens via a hormonal control dependence [19, 58]. Once circulating testosterone is transported into the prostate cells, it is converted by 5α-reductase into dihydrotestosterone (DHT), which in turn binds the androgen receptor (AR), dimerizes and translates into the cell nucleus. The nuclear AR then binds the response elements of androgen-regulated genes to trigger gene expression of secretory proteins or enzymes to be secreted into the semen [19, 39].  

In addition to hormonal control, the prostatic secretions also underlay neural control [19, 59]. It is noteworthy that the components of the prostate including both epithelial and stromal derived tissues are innervated by the autonomic nervous system [19, 60].

As has been widely reported and documented, prostate fluid is rich in enzymes, proteins and minerals that are essential for the protection and nutrition of sperm [60, 61]. Due to sexual arousal, the prostate can push prostate fluid through the ducts into the urethra, where it mixes with sperm and other fluids and is then ejaculated as semen [62, 63]. The most common prostatic fluid components secretions include Zinc, citric acid, Sodium, potassium, calcium, magnesium and chloride, spermine, prostaglandins, Cholesterol and lipids, immunoglobulins, acid phosphatase, and prostate specific antigen (PSA) as well as extracellular vesicles [64, 65].

The highest concentration of zinc has been reported in prostate gland, when compared he other organs within human body [64, 65]. Therefore, the prostatic secretions are the main source for higher concentration of zinc in the seminal plasma that is estimated to be nearly 100-fold higher than those in serum [66, 67] [66].  In addition to its function as an antibacterial factor both in seminal plasma and in non-ejaculatory prostate secretions as well, zinc is also thought to play an important role in the development of normal testes, spermatogenesis and sperm motility. [68]. Furthermore, the zinc in seminal plasma has been reported to function as a cell membrane and nuclear chromatin stabilizer in spermatozoa [69].

The binding of zinc to semenogelin I and II in seminal plasma is also essential for the regulation of semen liquefaction [70, 71]. Semenogelin is a protein reported to be involved in the formation of a gel matrix that coats ejaculated sperm to prevent capacitation [72, 73].  Citric acid is also one of the secretory products of the prostate, which is characterized by its ability to bind metal ions. It is also one of the main anions in human seminal plasma [74, 75]. Although higher concentrations of calcium and magnesium have been found in seminal plasma compared to other body fluids, the contribution of the prostate to the increase in these anions is not reported. [78]. However, variations in sodium, potassium, calcium, magnesium and chloride concentrations in prostate fluid have been reported to be closely associated with health or disease status [65, 78]. This is partly a passive response to varying amounts of citrate secretion in the prostate fluid in the disease state

Spermine is one of the polyamines that is discovered in semen [79, 80]. Spermine is one of the components of semen and its concentration in the prostate gland is the highest spermine concentration in the human body [81, 82]. It is also the main source of polyamine in seminal plasma. [83, 84].. The enzyme ornithine decarboxylase (ODC) is characterized by its ability to serve as a rate-limiting enzyme in the synthesis of polyamines and spermine [85]. ODC expression is increased in benign prostatic hyperplasia (BPH) tissue, suggesting a possible role of polyamines in the pathogenesis of this disease [86, 87].

Although prostaglandins are present throughout the body, the highest concentrations of prostaglandins in the human body have been found in the seminal vesicles [88, 89]. Prostaglandins are characterized by their potent physiological effects on men, particularly those involved in erection, ejaculation, sperm motility, and contractions of the testes and penis [90, 91]. While the deposition of prostaglandins in seminal fluid in the vagina has been reported to affect cervical mucus, vaginal secretions and uterine contractions [92]. 

The main source of the lipid fraction in seminal plasma comes from the prostate [93, 94]. The most common components of the lipid fraction in semen include cholesterol and phospholipids [95]. These phospholipids are composed of 44% sphingomyelin, 12.3% ethanolamine plasmalogen and 11.2% phosphatidylserine [96, 97]. While the ratio of cholesterol to phospholipids is considered important to protect sperm from temperature and environmental shocks, [98, 99].

The presence of immunoglobulins including IgG and IgA as well as IgM in seminal plasma at lower concentrations than those examined in blood [100, 101].

While the IgM concentration was lower in seminal plasma, when compared to those of IgG and IgA [102, 103].

The acid phosphatase activity is one of the components of the seminal fluid higher in the prostate than those reported in any other tissue [105, 106]. Prostatic acid phosphatase is a glycoprotein dimer, whose substrate in the seminal plasma is the phosphorylcholine phosphate that can rapidly be hydrolyzed by the acid phosphatase [106, 107]. Overexpression of acid phosphatase is observed in cancerous prostate tissues. For example, about 95% of patients with prostate cancer are characterized by high expression of acid phosphatase, especially in patients with metastases in the bones [107, 108]. Accordingly, acid phosphatase is considered an important serum marker for screening and staging of prostate cancer, even if it is replaced by the more specific prostate-specific antigen. (PSA).

PSA is a glycoprotein that acts as a serine protease and is produced almost exclusively in epithelial cells of the prostate [109, 110]. PSA is involved in the dissolution of the seminal clot and uses semenogelin as a substrate [111]. The clinical utilization of PSA as an important serum marker for prostate cancer has been established [106, 112]. PSA is an organ-specific rather than cancer-specific marker that is present in the serum of patients with BPH or prostatitis [113, 114].

As part of the components of seminal fluid, extracellular vehicles (EVs) are lipid bilayer nanovesicles, typically ranging in size from 30 to 1000 nm in diameter [115]. EVs serve as transporter of a wide range of molecules, including proteins, nucleic acids, metabolites and lipids [116,117]. Seminal plasma contains comparatively more EVs compared to those circulated in blood or cerebrospinal fluid [115, 116].

The main source of EVs in semen are the cells of the testes, epididymis and vas deferens, as well as the cells of the male accessory glands, including the prostate and seminal vesicles. [116]. Like EVs identified in other body fluids and containing a heterogeneous population characterized by their variable size, shape and composition [117, 118]. Semen EVs have the potential to bind to and exchange active molecules with mature sperm and endometrial epithelial cells and are thereby involved in the regulation of several reproductive processes. [116]. Furthermore, it has been reported that the cargo load of sEVs differs between normozoospermic and non-normozoospermic men. [119]. Seminal EVs would also facilitate the safe transit of spermatozoa in the female genital tract by regulating the uterine immune response [49, 116]. Taken together, these findings suggest that semen EVs may be involved in fertility and that some biomolecules encapsulated in EVs could be considered as biomarkers of male fertility. However, among the molecules encapsulated in sEVs, none are currently recognized as biomarkers for male fertility. To determine how far science has advanced in this field, the present systematic review aimed to summarize existing published evidence on the association of EVs with fertility and to identify candidate EV molecules that could function as biomarkers of fertility or infertility.

 Prostasomes are EVs that is known as macrovesicles that can be secreted the acinar epithelial cells of prostate gland [57, 120]. Prosteosomes are an intracellular vesicles  that are mostly found inside large storage vesicles  Intracellular macrovesicles are similar to the multivesicular bodies of the late endosomal origin [57]. Therefore, it was assumed that prostasomes play an important role in intercellular communication, especially by directly interacting between the immotile acinar cells of the prostate and the motile sperm. In addition to the transfer membrane components Prostasomes can also transfer genetic material to sperm [121]. Once the proteases are released from the secretory epithelial cells of the prostate gland and become part of the sperm, they can interact with the sperm membrane and eventually fuse with it. [57, 120], thus transferring molecules from prostatic secretory cell to the spermatozoon has been reported to be mediated by prosteosomes [57, 120].

Accumulated evidence suggests activation of the renin-angiotensin system (RAS) in the human prostate [122-124]. The expression and cellular localization of angiotensin II (Ang II) and AT1 receptor proteins were analyzed in normal human prostate and benign prostatic hyperplasia (BPH). As has been reported, increased expression of Ang II and significant reduction of the AT1 receptor are common in prostate hyperplasia are common [122-124]. While in the normal prostate the immunoreactivity was restricted to the basal layer of the epithelium, the AT1 receptor was predominantly localized in both the stroma and the smooth muscle of the prostatic vessels [122, 123]. The downregulation of the AT1 receptor together with increased tissue levels of Ang II as well as increased angiotensin-converting enzyme (ACE) at both mRNA and protein levels in BPH [125, 126], suggesting that RAS activation in BPH depends on its cellular localization and function. The role of Ang II and AT1 receptors in the regulation of prostate cancer has been demonstrated in several studies [127, 128]. The observation suggests clinical relevance to inhibition of RAS activation in the treatment and prevention of PCa [129-131]. Several studies have also shown that angiotensin II receptor blockers (ARBs) have the potential to reduce elevated PSA levels in patients without PCa and significantly reduce the risk of PCa. [131, 132]. .]has been added to chapter 2

References

  1. See lines: The following references [

Liu, A. Y.; True, L. D. Characterization of prostate cell types by CD cell surface molecules. Am J Pathol 2002160 (1), 37-43. DOI: 10.1016/S0002-9440(10)64346-5.

Samanta, L.; Parida, R.; Dias, T. R.; Agarwal, A. The enigmatic seminal plasma: a proteomics insight from ejaculation to fertilization. Reprod Biol Endocrinol 201816 (1), 41. DOI: 10.1186/s12958-018-0358-6.

Rodríguez-Martínez, H.; Kvist, U.; Ernerudh, J.; Sanz, L.; Calvete, J. J. Seminal plasma proteins: what role do they play? Am J Reprod Immunol 201166 Suppl 1, 11-22. DOI: 10.1111/j.1600-0897.2011.01033.x..

Ahmadi, H.; Csabai, T.; Gorgey, E.; Rashidiani, S.; Parhizkar, F.; Aghebati-Maleki, L. Composition and effects of seminal plasma in the female reproductive tracts on implantation of human embryos. Biomed Pharmacother 2022151, 113065. DOI: 10.1016/j.biopha.2022.113065.

Szczykutowicz, J.; KaÅ‚uża, A.; Kaźmierowska-Niemczuk, M.; Ferens-Sieczkowska, M. The Potential Role of Seminal Plasma in the Fertilization Outcomes. Biomed Res Int 20192019, 5397804. DOI: 10.1155/2019/5397804.

Dai, P.; Zou, M.; Cai, Z.; Zeng, X.; Zhang, X.; Liang, M. pH Homeodynamics and Male Fertility: A Coordinated Regulation of Acid-Based Balance during Sperm Journey to Fertilization. Biomolecules 202414 (6). DOI: 10.3390/biom14060685.

Juyena, N. S.; Stelletta, C. Seminal plasma: an essential attribute to spermatozoa. J Androl 201233 (4), 536-551. DOI: 10.2164/jandrol.110.012583.

Suarez, S. S.; Pacey, A. A. Sperm transport in the female reproductive tract. Hum Reprod Update 200612 (1), 23-37. DOI: 10.1093/humupd/dmi047.

Hawk, H. W. Sperm survival and transport in the female reproductive tract. J Dairy Sci 198366 (12), 2645-2660. DOI: 10.3168/jds.S0022-0302(83)82138-9.

Witkin, S. S.; Zelikovsky, G.; Bongiovanni, A. M.; Good, R. A.; Day, N. K. IgA antibody to spermatozoa in seminal and prostatic fluids of a subfertile man. JAMA 1982247 (7), 1014-1015.

Ronquist, G. Prostasomes are mediators of intercellular communication: from basic research to clinical implications. J Intern Med 2012271 (4), 400-413. DOI: 10.1111/j.1365-2796.2011.02487.x.  .

Vitku, J.; Kolatorova, L.; Hampl, R. Occurrence and reproductive roles of hormones in seminal plasma. Basic Clin Androl 201727, 19. DOI: 10.1186/s12610-017-0062-y.

Hernández-Aguilar, M. E.; Serrano, M. K.; Pérez, F.; Aranda-Abreu, G. E.; Sanchez, V.; Mateos, A.; Manzo, J.; Rojas-Durán, F.; Cruz-Gomez, Y.; Herrera-Covarrubias, D. Quantification of neural and hormonal receptors at the prostate of long-term sexual behaving male rats after lesion of pelvic and hypogastric nerves. Physiol Behav 2020222, 112915. DOI: 10.1016/j.physbeh.2020.112915.

White, C. W.; Xie, J. H.; Ventura, S. Age-related changes in the innervation of the prostate gland: implications for prostate cancer initiation and progression. Organogenesis 20139 (3), 206-215. DOI: 10.4161/org.24843.

Fontana, L.; Sirchia, S. M.; Pesenti, C.; Colpi, G. M.; Miozzo, M. R. Non-invasive biomarkers for sperm retrieval in non-obstructive patients: a comprehensive review. Front Endocrinol (Lausanne) 202415, 1349000. DOI: 10.3389/fendo.2024.1349000.

van Roijen, J. H.; Slob, A. K.; Gianotten, W. L.; Dohle, G. R.; van der Zon, A. T.; Vreeburg, J. T.; Weber, R. F. Sexual arousal and the quality of semen produced by masturbation. Hum Reprod 199611 (1), 147-151. DOI: 10.1093/oxfordjournals.humrep.a019008.

Pound, N.; Javed, M. H.; Ruberto, C.; Shaikh, M. A.; Del Valle, A. P. Duration of sexual arousal predicts semen parameters for masturbatory ejaculates. Physiol Behav 200276 (4-5), 685-689. DOI: 10.1016/s0031-9384(02)00803-x.

Lilja, H.; Abrahamsson, P. A. Three predominant proteins secreted by the human prostate gland. Prostate 198812 (1), 29-38. DOI: 10.1002/pros.2990120105.

Kavanagh, J. P. Sodium, potassium, calcium, magnesium, zinc, citrate and chloride content of human prostatic and seminal fluid. J Reprod Fertil 198575 (1), 35-41. DOI: 10.1530/jrf.0.0750035.

Costello, L. C.; Franklin, R. B. A comprehensive review of the role of zinc in normal prostate function and metabolism; and its implications in prostate cancer. Arch Biochem Biophys 2016611, 100-112. DOI: 10.1016/j.abb.2016.04.014.

Vashisht, A.; Gahlay, G. K. Understanding seminal plasma in male infertility: emerging markers and their implications. Andrology 202412 (5), 1058-1077. DOI: 10.1111/andr.13563.

Madding, C. I.; Jacob, M.; Ramsay, V. P.; Sokol, R. Z. Serum and semen zinc levels in normozoospermic and oligozoospermic men. Ann Nutr Metab 198630 (4), 213-218. DOI: 10.1159/000177196.

Lin, Y. C.; Chang, T. C.; Tseng, Y. J.; Lin, Y. L.; Huang, F. J.; Kung, F. T.; Chang, S. Y. Seminal plasma zinc levels and sperm motion characteristics in infertile samples. Chang Gung Med J 200023 (5), 260-266..

Johnson, R. D.; Hubscher, C. H. Brainstem microstimulation differentially inhibits pudendal motoneuron reflex inputs. Neuroreport 19989 (2), 341-345. DOI: 10.1097/00001756-199801260-00030.

Johnson, R. D.; Hubscher, C. H. Brainstem microstimulation differentially inhibits pudendal motoneuron reflex inputs. Neuroreport 19989 (2), 341-345. DOI: 10.1097/00001756-199801260-00030.

Edström, A. M.; Malm, J.; Frohm, B.; Martellini, J. A.; Giwercman, A.; Mörgelin, M.; Cole, A. M.; Sørensen, O. E. The major bactericidal activity of human seminal plasma is zinc-dependent and derived from fragmentation of the semenogelins. J Immunol 2008181 (5), 3413-3421. DOI: 10.4049/jimmunol.181.5.3413.

de Lamirande, E.; Yoshida, K.; Yoshiike, T. M.; Iwamoto, T.; Gagnon, C. Semenogelin, the main protein of semen coagulum, inhibits human sperm capacitation by interfering with the superoxide anion generated during this process. J Androl 200122 (4), 672-679

Sakaguchi, D.; Miyado, K.; Iwamoto, T.; Okada, H.; Yoshida, K.; Kang, W.; Suzuki, M.; Yoshida, M.; Kawano, N. Human Semenogelin 1 Promotes Sperm Survival in the Mouse Female Reproductive Tract. Int J Mol Sci 202021 (11). DOI: 10.3390/ijms21113961. 

Braude, P. R.; Ross, L. D.; Bolton, V. N.; Ockenden, K. Retrograde ejaculation: a systematic approach to non-invasive recovery of spermatozoa from post-ejaculatory urine for artificial insemination. Br J Obstet Gynaecol 198794 (1), 76-83. DOI: 10.1111/j.1471-0528.1987.tb02257.x.

Darbandi, M.; Darbandi, S.; Agarwal, A.; Sengupta, P.; Durairajanayagam, D.; Henkel, R.; Sadeghi, M. R. Reactive oxygen species and male reproductive hormones. Reprod Biol Endocrinol 201816 (1), 87. DOI: 10.1186/s12958-018-0406-2.

Gałęska, E.; WrzeciÅ„ska, M.; Kowalczyk, A.; Araujo, J. P. Reproductive Consequences of Electrolyte Disturbances in Domestic Animals. Biology (Basel) 202211 (7). DOI: 10.3390/biology11071006.

Asare, G. A.; Ngala, R. A.; Afriyie, D.; Adjei, S.; Nyarko, A.; Anang-Quartey, Y.; Asiedu, B.; Doku, D.; Amoah, B. Y.; Bentum, K.; et al. Calcium - Magnesium imbalance implicated in benign prostatic hyperplasia and restoration by a phytotherapeutic drug - Croton membranaceus Müll.Arg. BMC Complement Altern Med 201717 (1), 152. DOI: 10.1186/s12906-017-1663-x.

Coffino, P. Polyamines in spermiogenesis: not now, darling. Proc Natl Acad Sci U S A 200097 (9), 4421-4423. DOI: 10.1073/pnas.97.9.4421

Sagar, N. A.; Tarafdar, S.; Agarwal, S.; Tarafdar, A.; Sharma, S. Polyamines: Functions, Metabolism, and Role in Human Disease Management. Med Sci (Basel) 20219 (2). DOI: 10.3390/medsci9020044.

Peng, J.; Yang, L.; Liu, L.; Zhou, R.; Liu, J.; Li, N.; Li, L.; Jiang, Y.; Liu, Y.; Zhu, Z.; et al. Safety and Effectiveness of Dapoxetine On Demand in Chinese Men With Premature Ejaculation: Results of a Multicenter, Prospective, Open-Label Phase IV Study. Sex Med 20219 (2), 100296. DOI: 10.1016/j.esxm.2020.100296..

Prasher, P.; Sharma, M.; Singh, S. K.; Gulati, M.; Chellappan, D. K.; Rajput, R.; Gupta, G.; Ydyrys, A.; Kulbayeva, M.; Abdull Razis, A. F.; et al. Spermidine as a promising anticancer agent: Recent advances and newer insights on its molecular mechanisms. Front Chem 202311, 1164477. DOI: 10.3389/fchem.2023.1164477.

Lefèvre, P. L.; Palin, M. F.; Murphy, B. D. Polyamines on the reproductive landscape. Endocr Rev 201132 (5), 694-712. DOI: 10.1210/er.2011-0012.

Ploskonos, M. V.; Zulbalaeva, D. F.; Kurbangalieva, N. R.; Ripp, S. V.; Neborak, E. V.; Blagonravov, M. L.; Syatkin, S. P.; Sungrapova, K.; Hilal, A. Assessing the biological effects of microwave irradiation on human semen. Biomed Rep 202216 (5), 38. DOI: 10.3892/br.2022.1521.

Kapfhamer, D.; McKenna, J.; Yoon, C. J.; Murray-Stewart, T.; Casero, R. A.; Gambello, M. J. Ornithine decarboxylase, the rate-limiting enzyme of polyamine synthesis, modifies brain pathology in a mouse model of tuberous sclerosis complex. Hum Mol Genet 202029 (14), 2395-2407. DOI: 10.1093/hmg/ddaa121.

Liu, X.; Wang, L.; Lin, Y.; Teng, Q.; Zhao, C.; Hu, H.; Chi, W. Ornithine decarboxylase activity and its gene expression are increased in benign hyperplastic prostate. Prostate 200043 (2), 83-87. DOI: 10.1002/(sici)1097-0045(20000501)43:2<83::aid-pros2>3.0.co;2-o.

Shukla-Dave, A.; Castillo-Martin, M.; Chen, M.; Lobo, J.; Gladoun, N.; Collazo-Lorduy, A.; Khan, F. M.; Ponomarev, V.; Yi, Z.; Zhang, W.; et al. Ornithine Decarboxylase Is Sufficient for Prostate Tumorigenesis via Androgen Receptor Signaling. Am J Pathol 2016186 (12), 3131-3145. DOI: 10.1016/j.ajpath.2016.08.021..

Bach, D.; Walker, H. How important are prostaglandins in the urology of man? Urol Int 198237 (3), 160-171. DOI: 10.1159/000280813.

Kelly, R. W. Prostaglandin synthesis in the male and female reproductive tract. J Reprod Fertil 198162 (1), 293-304. DOI: 10.1530/jrf.0.0620293.

Song, Y.; Mao, C.; Zhong, Q.; Zhang, R.; Jiang, D.; Sun, X. Role of hydrogen sulfide in the male reproductive system. Front Endocrinol (Lausanne) 202415, 1377090. DOI: 10.3389/fendo.2024.1377090.

Semet, M.; Paci, M.; Saïas-Magnan, J.; Metzler-Guillemain, C.; Boissier, R.; Lejeune, H.; Perrin, J. The impact of drugs on male fertility: a review. Andrology 20175 (4), 640-663. DOI: 10.1111/andr.12366.

Fair, S.; Meade, K. G.; Reynaud, K.; Druart, X.; de Graaf, S. P. The biological mechanisms regulating sperm selection by the ovine cervix. Reproduction 2019158 (1), R1-R13. DOI: 10.1530/REP-18-0595. 

Arienti, G.; Saccardi, C.; Carlini, E.; Verdacchi, R.; Palmerini, C. A. Distribution of lipid and protein in human semen fractions. Clin Chim Acta 1999289 (1-2), 111-120. DOI: 10.1016/s0009-8981(99)00169-2.

Girouard, J.; Frenette, G.; Sullivan, R. Seminal plasma proteins regulate the association of lipids and proteins within detergent-resistant membrane domains of bovine spermatozoa. Biol Reprod 200878 (5), 921-931. DOI: 10.1095/biolreprod.107.066514.

Gautier, C.; Aurich, C. "Fine feathers make fine birds" - The mammalian sperm plasma membrane lipid composition and effects on assisted reproduction. Anim Reprod Sci 2022246, 106884. DOI: 10.1016/j.anireprosci.2021.106884.

Lenzi, A.; Picardo, M.; Gandini, L.; Dondero, F. Lipids of the sperm plasma membrane: from polyunsaturated fatty acids considered as markers of sperm function to possible scavenger therapy. Hum Reprod Update 19962 (3), 246-256. DOI: 10.1093/humupd/2.3.246.

Hinkovska, V. T.; Dimitrov, G. P.; Koumanov, K. S. Phospholipid composition and phospholipid asymmetry of ram spermatozoa plasma membranes. Int J Biochem 198618 (12), 1115-1121. DOI: 10.1016/0020-711x(86)90085-6.

Carro, M. L. M.; Peñalva, D. A.; Antollini, S. S.; Hozbor, F. A.; Buschiazzo, J. Cholesterol and desmosterol incorporation into ram sperm membrane before cryopreservation: Effects on membrane biophysical properties and sperm quality. Biochim Biophys Acta Biomembr 20201862 (9), 183357. DOI: 10.1016/j.bbamem.2020.183357.

Müller, K.; Müller, P.; Pincemy, G.; Kurz, A.; Labbe, C. Characterization of sperm plasma membrane properties after cholesterol modification: consequences for cryopreservation of rainbow trout spermatozoa. Biol Reprod 200878 (3), 390-399. DOI: 10.1095/biolreprod.107.064253.

Luckas, M. J.; Buckett, W. M.; Aird, I. A.; Johnson, P. M.; Lewis-Jones, D. I. Seminal plasma immunoglobulin concentrations in autoimmune male subfertility. J Reprod Immunol 199837 (2), 171-180. DOI: 10.1016/s0165-0378(97)00080-6.

Silva, J. A. F.; Biancardi, M. F.; Stach-Machado, D. R.; Reis, L. O.; Sant'Anna, O. A.; Carvalho, H. F. The origin of prostate gland-secreted IgA and IgG. Sci Rep 20177 (1), 16488. DOI: 10.1038/s41598-017-16717-3.

Pillay, T.; Sobia, P.; Olivier, A. J.; Narain, K.; Liebenberg, L. J. P.; Ngcapu, S.; Mhlongo, M.; Passmore, J. S.; Baxter, C.; Archary, D. Semen IgM, IgG1, and IgG3 Differentially Associate With Pro-Inflammatory Cytokines in HIV-Infected Men. Front Immunol 20189, 3141. DOI: 10.3389/fimmu.2018.03141.

Donat, H.; Schiller, S. [Immunoglobulins in human seminal plasma]. Andrologia 198315 (4), 369-373.

[104] Sirigu, P.; Perra, M. T.; Turno, F. Immunohistochemical study of secretory IgA in the human male reproductive tract. Andrologia 199527 (6), 335-339. DOI: 10.1111/j.1439-0272.1995.tb01368.x.

Muniyan, S.; Chaturvedi, N. K.; Dwyer, J. G.; Lagrange, C. A.; Chaney, W. G.; Lin, M. F. Human prostatic acid phosphatase: structure, function and regulation. Int J Mol Sci 201314 (5), 10438-10464. DOI: 10.3390/ijms140510438.

Hanley, P. J. Elusive physiological role of prostatic acid phosphatase (PAP): generation of choline for sperm motility via auto-and paracrine cholinergic signaling. Front Physiol 202314, 1327769. DOI: 10.3389/fphys.2023.1327769.

Hanley, P. J. Elusive physiological role of prostatic acid phosphatase (PAP): generation of choline for sperm motility via auto-and paracrine cholinergic signaling. Front Physiol 202314, 1327769. DOI: 10.3389/fphys.2023.1327769.

Kong, H. Y.; Byun, J. Emerging roles of human prostatic Acid phosphatase. Biomol Ther (Seoul) 201321 (1), 10-20. DOI: 10.4062/biomolther.2012.095.

Heinrich, D.; Bruland, Ø.; Guise, T. A.; Suzuki, H.; Sartor, O. Alkaline phosphatase in metastatic castration-resistant prostate cancer: reassessment of an older biomarker. Future Oncol 201814 (24), 2543-2556. DOI: 10.2217/fon-2018-0087.

el-Shirbiny, A. M. Prostatic specific antigen. Adv Clin Chem 199431, 99-133. DOI: 10.1016/s0065-2423(08)60334-0.

Sutkowski, D. M.; Goode, R. L.; Baniel, J.; Teater, C.; Cohen, P.; McNulty, A. M.; Hsiung, H. M.; Becker, G. W.; Neubauer, B. L. Growth regulation of prostatic stromal cells by prostate-specific antigen. J Natl Cancer Inst 199991 (19), 1663-1669. DOI: 10.1093/jnci/91.19.1663.

Robert, M.; Gibbs, B. F.; Jacobson, E.; Gagnon, C. Characterization of prostate-specific antigen proteolytic activity on its major physiological substrate, the sperm motility inhibitor precursor/semenogelin I. Biochemistry 199736 (13), 3811-3819. DOI: 10.1021/bi9626158.

Gersuk, G. M.; Corey, M. J.; Corey, E.; Stray, J. E.; Kawasaki, G. H.; Vessella, R. L. High-affinity peptide ligands to prostate-specific antigen identified by polysome selection. Biochem Biophys Res Commun 1997232 (2), 578-582. DOI: 10.1006/bbrc.1997.6331.

Sarwar, S.; Adil, M. A.; Nyamath, P.; Ishaq, M. Biomarkers of Prostatic Cancer: An Attempt to Categorize Patients into Prostatic Carcinoma, Benign Prostatic Hyperplasia, or Prostatitis Based on Serum Prostate Specific Antigen, Prostatic Acid Phosphatase, Calcium, and Phosphorus. Prostate Cancer 20172017, 5687212. DOI: 10.1155/2017/5687212...

Ergun, O.; Capar, E.; Goger, Y. E.; Ergun, A. G. Can expressed prostatic secretions effect prostate biopsy decision of urologist? Int Braz J Urol 201945 (2), 246-252. DOI: 10.1590/S1677-5538.IBJU.2018.0292.

[115] Ramesh, D.; Bakkannavar, S.; Bhat, V. R.; Sharan, K. Extracellular vesicles as novel drug delivery systems to target cancer and other diseases: Recent advancements and future perspectives. F1000Res 202312, 329. DOI: 10.12688/f1000research.132186.1.

[116 ] Parra, A.; Padilla, L.; Lucas, X.; Rodriguez-Martinez, H.; Barranco, I.; Roca, J. Seminal Extracellular Vesicles and Their Involvement in Male (In)Fertility: A Systematic Review. Int J Mol Sci 202324 (5). DOI: 10.3390/ijms24054818.

[117] Ayaz, A.; Houle, E.; Pilsner, J. R. Extracellular vesicle cargo of the male reproductive tract and the paternal preconception environment. Syst Biol Reprod Med 202167 (2), 103-111. DOI: 10.1080/19396368.2020.1867665.

Fernandez-Fuertes, B. Review: The role of male reproductive tract secretions in ruminant fertility. Animal 202317 Suppl 1, 100773. DOI: 10.1016/j.animal.2023.100773.

Roy, R.; Lorca, C.; Mulet, M.; Sánchez Milán, J. A.; Baratas, A.; de la Casa, M.; Espinet, C.; Serra, A.; Gallart-Palau, X. Altered ureido protein modification profiles in seminal plasma extracellular vesicles of non-normozoospermic men. Front Endocrinol (Lausanne) 202314, 1113824. DOI: 10.3389/fendo.2023.1113824.

Park, K. H.; Kim, B. J.; Kang, J.; Nam, T. S.; Lim, J. M.; Kim, H. T.; Park, J. K.; Kim, Y. G.; Chae, S. W.; Kim, U. H. Ca2+ signaling tools acquired from prostasomes are required for progesterone-induced sperm motility. Sci Signal 20114 (173), ra31. DOI: 10.1126/scisignal.2001595.

Fabiani, R.; Johansson, L.; Lundkvist, O.; Ronquist, G. Enhanced recruitment of motile spermatozoa by prostasome inclusion in swim-up medium. Hum Reprod 19949 (8), 1485-1489. DOI: 10.1093/oxfordjournals.humrep.a138735.

Dinh, D. T.; Frauman, A. G.; Sourial, M.; Casley, D. J.; Johnston, C. I.; Fabiani, M. E. Identification, distribution, and expression of angiotensin II receptors in the normal human prostate and benign prostatic hyperplasia. Endocrinology 2001142 (3), 1349-1356. DOI: 10.1210/endo.142.3.8020.

Dinh, D. T.; Frauman, A. G.; Somers, G. R.; Ohishi, M.; Zhou, J.; Casley, D. J.; Johnston, C. I.; Fabiani, M. E. Evidence for activation of the renin-angiotensin system in the human prostate: increased angiotensin II and reduced AT(1) receptor expression in benign prostatic hyperplasia. J Pathol 2002196 (2), 213-219. DOI: 10.1002/path.1021.

Hassani, B.; Attar, Z.; Firouzabadi, N. The renin-angiotensin-aldosterone system (RAAS) signaling pathways and cancer: foes versus allies. Cancer Cell Int 202323 (1), 254. DOI: 10.1186/s12935-023-03080-9.

Nassis, L.; Frauman, A. G.; Ohishi, M.; Zhuo, J.; Casley, D. J.; Johnston, C. I.; Fabiani, M. E. Localization of angiotensin-converting enzyme in the human prostate: pathological expression in benign prostatic hyperplasia. J Pathol 2001195 (5), 571-579. DOI: 10.1002/path.999.

Kaplan, S. A. Localization of angiotensin-converting enzyme in the human prostate: pathological expression in benign prostatic hyperplasia. J Urol 2002168 (4 Pt 1), 1659.

Uemura, H.; Kubota, Y. [Role of renin-angiotensin system in prostate cancer]. Gan To Kagaku Ryoho 200936 (8), 1228-1233.

Chow, L.; Rezmann, L.; Catt, K. J.; Louis, W. J.; Frauman, A. G.; Nahmias, C.; Louis, S. N. Role of the renin-angiotensin system in prostate cancer. Mol Cell Endocrinol 2009302 (2), 219-229. DOI: 10.1016/j.mce.2008.08.032.

DomiÅ„ska, K.; OkÅ‚a, P.; Kowalska, K.; Habrowska-GórczyÅ„ska, D. E.; Urbanek, K. A.; OchÄ™dalski, T.; Piastowska-Ciesielska, A. W. Angiotensin 1-7 modulates molecular and cellular processes central to the pathogenesis of prostate cancer. Sci Rep 20188 (1), 15772. DOI: 10.1038/s41598-018-34049-8.

DomiÅ„ska, K.; OkÅ‚a, P.; Kowalska, K.; Habrowska-GórczyÅ„ska, D. E.; Urbanek, K. A.; OchÄ™dalski, T.; Piastowska-Ciesielska, A. W. Angiotensin 1-7 modulates molecular and cellular processes central to the pathogenesis of prostate cancer. Sci Rep 20188 (1), 15772. DOI: 10.1038/s41598-018-34049-8.

Iheanacho, C. O.; Enechukwu, O. H. Role of antihypertensive medicines in prostate cancer: a systematic review. BMC Cancer 202424 (1), 542. DOI: 10.1186/s12885-024-12218-5. ] have been added to references section

Comment 2: Minor comment: page 20/30: Reference 81 is incomplete 

Authors’response: Thank you very much for comment. We apologize for the mistake. Accordingly, we replaced the incomplete reference with the corrected one

See lines: 1263-164; the incomplete reference [, B. The role of oxytocin in male and female reproductive behavior. Eur J Pharmacol 2015, 753, 209-228. DOI: 10.1016/j.ejphar.2014.07.045.] by the  corrected one [ Veening, J. G.; de Jong, T. R.; Waldinger, M. D.; Korte, S. M.; Olivier, B. The role of oxytocin in male and female reproductive behavior. Eur J Pharmacol 2015753, 209-228. DOI: 10.1016/j.ejphar.2014.07.045.]

Reviewer 3 Report

Comments and Suggestions for Authors

Dear Editor

  1. The title of the review is misleading. The appropriate title should be: How frequent ejaculations reduce the risk of prostate cancer?
  2. The abstract basically says nothing. Pressure space is taken by anatomical and physiological explanations that are not relevant to the subject.
  3. Then comes long and out of the subject endocrinological description.
  4. The connection between frequent ejaculation and reduced PC risk is not a general knowledge and one would expect a paragraph describing this.
  5. Some of the text is underlined without any obvious reason.
  6. It is only on page 14 that the authors finally get to the point. Chapter 7 is well written, and its summary should appear in the abstract.

Author Response

Author response to Reviewer 2

Dear Editor,

Enclosed find please our response to the valuable comment of reviewer 02

Reviewer 3

Comments and Suggestions for Authors

Commen 1: The title of the review is misleading. The appropriate title should be: How frequent ejaculations reduce the risk of prostate cancer?

Authors’response: Thank you very much for your comment, As required, we changed the title to be more suitable.

See lines: 2-3; The current title [Mechanisms Regulating the Reduction of Prostate Cancer Risk ] has been modified and replaced by the following title [Reduction of Prostate Cancer Risk: Role of Frequent Ejaculation-Associated Mechanisms]

Comment 2: The abstract basically says nothing. Pressure space is taken by anatomical and physiological explanations that are not relevant to the subject.

Authors’response:

Comment 3: Then comes long and out of the subject endocrinological description.

Authors’response: Thank you very much for your comment

Comment 4: The connection between frequent ejaculation and reduced PC risk is not general knowledge and one would expect a paragraph describing this.

Authors’response: Thank you very much for your valuable comment. Accordingly, we modified the title to be more suitable and also, we made significant modification and changes in the overall structure of the manuscript. Accordingly, the following modifications and changes have been made.

See lines: 2-3; The current title [Mechanisms Regulating the Reduction of Prostate Cancer Risk ] has been modified and replaced by the following title [Reduction of Prostate Cancer Risk: Role of Frequent Ejaculation-Associated Mechanisms]

See Lines:794-831; the following chapter  [8. Mechanisms of frequent ejaculation-induced reduction of prostate cancer risk The regulation of PCa risk induced by frequent ejaculation appears to be mediated by several mechanisms that have been proposed to explain the association between frequent ejaculation and reduced PCa [372, 373]. The higher the frequency of ejaculation, the more inhibited the metabolic switch from citrate secretion to citrate oxidation in the epithelial cells in the peripheral zone. One of the possible mechanisms by which frequent ejaculation may reduce the risk of tumor development in the prostate [373]. Also, frequent ejaculation is expected to reduce PCa risk via mechanism mediated by restore the development of intraluminal eosinophilic structures in the prostate, which are considered a sign of higher risk for PCa. [374, 375]. Another mechanism by which frequent ejaculation can reduce PCA risk is expected to be mediated by reducing psychological tension and suppressing the central sympathetic nervous system, resulting in a significant attenuation of the stimulation of prostate epithelial cell division. [376].

As is generally accepted, cancer is one of the most common diseases caused by environmental and external influences [377, 378]. The development of around 90% of cancers diagnosed worldwide is attributed mainly to harmful environmental chemicals [379]. Among these substances, several active harmful chemicals that are recognized as endocrine disruptors [380, 381]. These endocrine disruptors have common characteristics that are mostly associated with disruption of hormone regulation and action [382]. Endocrine disruptors act primarily via nuclear hormone receptors, such as estrogen and progesterone receptors [383, 384]. The mechanisms of endocrine disruptors are much more comprehensive than commonly reported [385, 386].

Like other organs, the prostate is a target of numerous environmentally harmful chemicals including endocrine disruptors that can induce multi-organ disease through mechanisms based on interaction with the cellular receptors or components [387, 388]. Therefore, the interference of these harmful chemicals with the action of hormones, which are involved in the different cellular functions of the prostate gland is expected to develop hormone-dependent malignancies [389, 390]. Ultimately, many reports suggested that endocrine disruptors have the potential to significantly alter the epigenomic landscape in cancers, including prostate cancer [391, 392]. Thus, frequent ejaculation is expected to protect the prostate by flushing out these harmful chemicals. Consequently, men who ejaculate more frequently can have healthy lifestyle habits harmful chemicals that can, in turn, reduce the risk of prostate cancer.

The stimulation of endocannabinoid release in human by masturbation to orgasm suggests an essential role for the frequent ejaculation in the reduction of prostate cancer via endogenous 2-arachidonoylglycerol (2-AG) -dependent mechanism [393]. Also, several studies demonstrated the ability of 2-AG to inhibit the invasion of androgen-independent prostate cancer PC-3 and DU-145) cell lines in vitro [393]. ] has been added to the main text.

See lines: 1748-1802; the following references [

Lozano-Lorca, M.; Olmedo-Requena, R.; Barrios-Rodríguez, R.; Jiménez-Pacheco, A.; Vázquez-Alonso, F.; Castillo-Bueno, H. M.; Rodríguez-Barranco, M.; Jiménez-Moleón, J. J. Ejaculation Frequency and Prostate Cancer: CAPLIFE Study. World J Mens Health 202341 (3), 724-733. DOI: 10.5534/wjmh.220216

Costello, L. C.; Franklin, R. B. The clinical relevance of the metabolism of prostate cancer; zinc and tumor suppression: connecting the dots. Mol Cancer 20065, 17. DOI: 10.1186/1476-4598-5-17.

Svatek, R. S.; Karam, J. A.; Rogers, T. E.; Shulman, M. J.; Margulis, V.; Benaim, E. A. Intraluminal crystalloids are highly associated with prostatic adenocarcinoma on concurrent biopsy specimens. Prostate Cancer Prostatic Dis 200710 (3), 279-282. DOI: 10.1038/sj.pcan.4500954.

Del Rosario, A. D.; Bui, H. X.; Abdulla, M.; Ross, J. S. Sulfur-rich prostatic intraluminal crystalloids: a surgical pathologic and electron probe x-ray microanalytic study. Hum Pathol 199324 (11), 1159-1167. DOI: 10.1016/0046-8177(93)90210-8.

Newman, H. F.; Reiss, H.; Northup, J. D. Physical basis of emission, ejaculation, and orgasm in the male. Urology 198219 (4), 341-350. DOI: 10.1016/0090-4295(82)90186-8.

Parsa, N. Environmental factors inducing human cancers. Iran J Public Health 201241 (11), 1-9.

Baillie-Hamilton, P. F. Chemical toxins: a hypothesis to explain the global obesity epidemic. J Altern Complement Med 20028 (2), 185-192. DOI: 10.1089/107555302317371479.

Madia, F.; Worth, A.; Whelan, M.; Corvi, R. Carcinogenicity assessment: Addressing the challenges of cancer and chemicals in the environment. Environ Int 2019128, 417-429. DOI: 10.1016/j.envint.2019.04.067.

Jensen, T. K.; Andersson, A. M.; Jørgensen, N.; Andersen, A. G.; Carlsen, E.; Petersen, J. H.; Skakkebaek, N. E. Body mass index in relation to semen quality and reproductive hormones among 1,558 Danish men. Fertil Steril 200482 (4), 863-870. DOI: 10.1016/j.fertnstert.2004.03.056.

Macdonald, A. A.; Stewart, A. W.; Farquhar, C. M. Body mass index in relation to semen quality and reproductive hormones in New Zealand men: a cross-sectional study in fertility clinics. Hum Reprod 201328 (12), 3178-3187. DOI: 10.1093/humrep/det379.

La Merrill, M. A.; Vandenberg, L. N.; Smith, M. T.; Goodson, W.; Browne, P.; Patisaul, H. B.; Guyton, K. Z.; Kortenkamp, A.; Cogliano, V. J.; Woodruff, T. J.; et al. Consensus on the key characteristics of endocrine-disrupting chemicals as a basis for hazard identification. Nat Rev Endocrinol 202016 (1), 45-57. DOI: 10.1038/s41574-019-0273-8.  

Lee, H. R.; Jeung, E. B.; Cho, M. H.; Kim, T. H.; Leung, P. C.; Choi, K. C. Molecular mechanism(s) of endocrine-disrupting chemicals and their potent oestrogenicity in diverse cells and tissues that express oestrogen receptors. J Cell Mol Med 201317 (1), 1-11. DOI: 10.1111/j.1582-4934.2012.01649.x.

Tabb, M. M.; Blumberg, B. New modes of action for endocrine-disrupting chemicals. Mol Endocrinol 200620 (3), 475-482. DOI: 10.1210/me.2004-0513..

Diamanti-Kandarakis, E.; Bourguignon, J. P.; Giudice, L. C.; Hauser, R.; Prins, G. S.; Soto, A. M.; Zoeller, R. T.; Gore, A. C. Endocrine-disrupting chemicals: an Endocrine Society scientific statement. Endocr Rev 200930 (4), 293-342. DOI: 10.1210/er.2009-0002.

Ankolkar, M.; Balasinor, N. H. Endocrine control of epigenetic mechanisms in male reproduction. Horm Mol Biol Clin Investig 201625 (1), 65-70. DOI: 10.1515/hmbci-2016-0007 

Corti, M.; Lorenzetti, S.; Ubaldi, A.; Zilli, R.; Marcoccia, D. Endocrine Disruptors and Prostate Cancer. Int J Mol Sci 202223 (3). DOI: 10.3390/ijms23031216.

Bleak, T. C.; Calaf, G. M. Breast and prostate glands affected by environmental substances (Review). Oncol Rep 202145 (4). DOI: 10.3892/or.2021.7971

Lacouture, A.; Lafront, C.; Peillex, C.; Pelletier, M.; Audet-Walsh, É. Impacts of endocrine-disrupting chemicals on prostate function and cancer. Environ Res 2022204 (Pt B), 112085. DOI: 10.1016/j.envres.2021.112085.

Prins, G. S. Endocrine disruptors and prostate cancer risk. Endocr Relat Cancer 200815 (3), 649-656. DOI: 10.1677/ERC-08-0043.

Lehle, J. D.; McCarrey, J. R. Differential susceptibility to endocrine disruptor-induced epimutagenesis. Environ Epigenet 20206 (1), dvaa016. DOI: 10.1093/eep/dvaa016.

Lang, I. A.; Galloway, T. S.; Scarlett, A.; Henley, W. E.; Depledge, M.; Wallace, R. B.; Melzer, D. Association of urinary bisphenol A concentration with medical disorders and laboratory abnormalities in adults. JAMA 2008300 (11), 1303-1310. DOI: 10.1001/jama.300.11.1303. 31, 41, 46].

Fuss, J.; Bindila, L.; Wiedemann, K.; Auer, M. K.; Briken, P.; Biedermann, S. V. Masturbation to Orgasm Stimulates the Release of the Endocannabinoid 2-Arachidonoylglycerol in Humans. J Sex Med 201714 (11), 1372-1379. DOI: 10.1016/j.jsxm.2017.09.016.

 Nithipatikom, K.; Endsley, M. P.; Isbell, M. A.; Falck, J. R.; Iwamoto, Y.; Hillard, C. J.; Campbell, W. B. 2-arachidonoylglycerol: a novel inhibitor of androgen-independent prostate cancer cell invasion. Cancer Res 200464 (24), 8826-8830. DOI: 10.1158/0008-5472.CAN-04-3136. . ] have been added to reference section

Comment 5: Some of the text is underlined without any obvious reason.

Authors’response: Authors’ response: Thank you very much for your comment. The underlined paragraphs were the authors ‘response to the comment of the editor. Accordingly, we underlined the changes that have been made in the manuscript.

Comment 6: It is only on page 14 that the authors finally get to the point. Chapter 7 is well written, and its summary should appear in the abstract.

Authors’response: Thank you very much for your valuable comment. As you suggested, we modified the abstract

See lines:34-45; the following text [Abstract: Prostate cancer (PCa) accounts for roughly 15% of diagnosed cancer among men, with disease incidence increasing worldwide. Age, family history and ethnicity, diet, physical activity, and chemoprevention all play a role in reducing PCa risk. Although frequent ejaculation has been reported to play a significant role in the reduction of PCa risk, the mechanisms regulating frequent ejaculation- mediated reduction of PCa risk are not well described. The prostate gland is part of the male reproductive system located inside the body. As an exocrine gland, the prostate is a multi-functional gland involved in reproductive aspects such as male ejaculation and orgasmic ecstasy, as well as plays key roles in the regulation of local and systemic concentrations of 5α-dihydrotestosterone. 5α-dihydrotestosterone, expressed in the prostate, is an important potent androgen which results from the conversion of the precursor testosterone via mechanism mediated by isoenzyme 5α- reductase-2. Recent evidence indicates that frequent ejaculation plays a potential role in reducing PCa risk. Accordingly, more frequent ejaculation in the absence of risky sexual behavior likewise improves PCa risk. In this review, we provide an insight into possible mechanisms regulating the impact of frequent ejaculation on reducing PCa risk. ] has been replaced by a new one [Abstract: Prostate cancer (PCa) accounts for roughly 15% of diagnosed cancer among men, with disease incidence increasing worldwide. Age, family history and ethnicity, diet, physical activity, and chemoprevention all play a role in reducing PCa risk. The prostate is an exocrine gland that is characterized by its multi-function involved in reproductive aspects such as male ejaculation and orgasmic ecstasy, as well as plays key roles in the regulation of local and systemic concentrations of 5α-dihydrotestosterone. The increase in androgen receptors at the ventral prostate is the first elevated response induced by copulation. The regulation of prostate growth and function is mediated by androgens-dependent mechanism. Binding 5-DHT binds to androgen receptors (AR) results in the formation of a 5α-DHT:AR complex.  The interaction of 5α-DHT:AR complex with the specific DNA enhancer element of androgen-regulated genes leading to the regulation of androgen-specific target genes to maintain prostate homeostasis. Consequently, ejaculation may play a significant role in the reduction of PCa risk. Thus, the frequent ejaculation in the absence of risky sexual behavior likewise is a possible approach or the prevention of PCa. In this review, we provide an insight into possible mechanisms regulating the impact of frequent ejaculation on reducing PCa risk.]  

Round 2

Reviewer 1 Report

Comments and Suggestions for Authors

The authors have significantly improved the manuscript!

Author Response

Authors‘response to Reviewer 1

Comments and Suggestions for Authors: The authors have significantly improved the manuscript!

Authors ‘response: Thank you very much for your comment

Reviewer 2 Report

Comments and Suggestions for Authors

This revised manuscript is sufficiently improved.

Please observe a small notice:

An error has slipped into the text in next last paragraph of Discussion (before References):

Prostasomes have quite accidentally been denoted  "prosteosomes"

Author Response

Authors ‘response to Reviewer 2

Comments and Suggestions for Authors: This revised manuscript is sufficiently improved. Please observe a small notice: An error has slipped into the text in next last paragraph of Discussion (before References): Prostasomes have quite accidentally been denoted  "prosteosomes

"

Auhors’response: Thank you very much for your valuable comment. We apologize for this mistake. We have made the required corrections.

See line:226, 228, 231, 234;   The word “ prosteosomes” has been corrected and replaced by the corrected word “ Prostasomes “

Reviewer 3 Report

Comments and Suggestions for Authors

I did not find the response to comment 2

Author Response

Authors’response to Reviewer 3

Comment 2: The abstract basically says nothing. Pressure space is taken by anatomical and physiological explanations that are not relevant to the subject.

Authors ‘response: Thank you very much for your comment, We apologize that we forget to mention the changes performed and modification of the abstract.

See lines:37-52; the following abstract [ Prostate cancer (PCa) accounts for roughly 15% of diagnosed cancer among men, with disease incidence increasing worldwide. Age, family history and ethnicity, diet, physical activity, and chemoprevention all play a role in reducing PCa risk. Although frequent ejaculation has been reported to play a significant role in the reduction of PCa risk, the mechanisms regulating frequent ejaculation- mediated reduction of PCa risk are not well described. The prostate gland is part of the male reproductive system located inside the body. As an exocrine gland, the prostate is a multi-functional gland involved in reproductive aspects such as male ejaculation and orgasmic ecstasy, as well as plays key roles in the regulation of local and systemic concentrations of 5α-dihydrotestosterone. 5α-dihydrotestosterone, expressed in the prostate, is an important potent androgen which results from the conversion of the precursor testosterone via mechanism mediated by isoenzyme 5α- reductase-2. Recent evidence indicates that frequent ejaculation plays a potential role in reducing PCa risk. Accordingly, more frequent ejaculation in the absence of risky sexual behavior likewise improves PCa risk. In this review, we provide an insight into possible mechanisms regulating the impact of frequent ejaculation on reducing PCa risk. ] has been modified and replaced by the following abstract[Abstract: Prostate cancer (PCa) accounts for roughly 15% of diagnosed cancer among men, with disease incidence increasing worldwide. Age, family history and ethnicity, diet, physical activity, and chemoprevention all play a role in reducing PCa risk. The prostate is an exocrine gland that is characterized by its multi-function involved in reproductive aspects such as male ejaculation and orgasmic ecstasy, as well as plays key roles in the regulation of local and systemic concentrations of 5α-dihydrotestosterone. The increase in androgen receptors at the ventral prostate is the first elevated response induced by copulation. The regulation of prostate growth and function is mediated by androgens-dependent mechanism. Binding 5-DHT binds to androgen receptors (AR) results in the formation of a 5α-DHT:AR complex.  The interaction of 5α-DHT:AR complex with the specific DNA enhancer element of androgen-regulated genes leading to the regulation of androgen-specific target genes to maintain prostate homeostasis. Consequently, ejaculation may play a significant role in the reduction of PCa risk. Thus, the frequent ejaculation in the absence of risky sexual behavior likewise is a possible approach or the prevention of PCa. In this review, we provide an insight into possible mechanisms regulating the impact of frequent ejaculation on reducing PCa risk.  ]
